# Transcriptome network analysis implicates CX3CR1-positive type 3 dendritic cells in non-infectious uveitis

**Sanne Hiddingh**[1,2†], **Aridaman Pandit**[2,3†], **Fleurieke Verhagen**[1,2,3†], **Rianne Rijken**[2,4], **Nila Hendrika Servaas**[2,4], **Rina CGK Wichers**[2,4], **Ninette H ten Dam-van Loon**[3], **Saskia M Imhof**[1,3], **Timothy RDJ Radstake**[5], **Joke H de Boer**[1,3], **Jonas JW Kuiper**[1,2,3]*

[1]Ophthalmo-Immunology, University Medical Center Utrecht, Utrecht University, Utrecht, Netherlands; [2]Center for Translational Immunology, University Medical Center Utrecht, Utrecht University, Utrecht, Netherlands; [3]Department of Ophthalmology, University Medical Center Utrecht, Utrecht University, Utrecht, Netherlands; [4]Department of Rheumatology & Clinical Immunology, University Medical Center Utrecht, Utrecht University, Utrecht, Netherlands; [5]University Medical Center Utrecht and Utrecht University, Utrecht, Netherlands

**\*For correspondence:**
j.j.w.kuiper@umcutrecht.nl

†These authors contributed equally to this work

## Abstract

**Background:** Type I interferons (IFNs) promote the expansion of subsets of CD1c+ conventional dendritic cells (CD1c+ DCs), but the molecular basis of CD1c+ DCs involvement in conditions not associated without elevated type I IFNs remains unclear.

**Methods:** We analyzed CD1c+ DCs from two cohorts of non-infectious uveitis patients and healthy donors using RNA-sequencing followed by high-dimensional flow cytometry to characterize the CD1c+ DC populations.

**Results:** We report that the CD1c+ DCs pool from patients with non-infectious uveitis is skewed toward a gene module with the chemokine receptor *CX3CR1* as the key hub gene. We confirmed these results in an independent case–control cohort and show that the disease-associated gene module is not mediated by type I IFNs. An analysis of peripheral blood using flow cytometry revealed that CX3CR1+ DC3s were diminished, whereas CX3CR1− DC3s were not. Stimulated CX3CR1+ DC3s secrete high levels of inflammatory cytokines, including TNF-alpha, and CX3CR1+ DC3 like cells can be detected in inflamed eyes of patients.

**Conclusions:** These results show that CX3CR1+ DC3s are implicated in non-infectious uveitis and can secrete proinflammatory mediators implicated in its pathophysiology.

**Funding:** The presented work is supported by UitZicht (project number #2014-4, #2019-10, and #2021-4). The funders had no role in the design, execution, interpretation, or writing of the study.

## Editor's evaluation

These findings are valuable to ocular immunologists who the study pathophysiologic mechanisms driving inflammation in human uveitis, and for future identification of novel therapeutic targets. The authors convincingly perform high dimensional multi-omic analysis of testing and replication cohorts, followed by characterization of a disease-specific cell type using comparative analysis with previously validated experimental datasets. The analysis will be of particular interest to basic and translational ocular immunologists, as well as dendritic cell biologists.

## Introduction

Non-infectious uveitis refers to a group of chronic inflammatory eye diseases that are among the leading causes of preventable vision loss in western countries (*Thorne et al., 2016*; *Suttorp-Schulten and Rothova, 1996*). Currently, little is known about the disease mechanism of non-infectious uveitis. A large body of mechanistic studies using experimental autoimmune uveitis (EAU) in rodents suggest that T cells play a role in non-infectious uveitis (*Lee et al., 2014*; *Caspi, 2010*). Human non-infectious uveitis is characterized by inflammation and T cells infiltrating the eyes through unknown mechanisms. The genetic association between non-infectious uveitis and *MHC*, and *ERAP1*, *ERAP2* genes indicates that antigen presentation is central to the etiology (*Kuiper and Venema, 2020*; *Huang et al., 2020*; *Kuiper et al., 2018*; *Márquez et al., 2017*). A key antigen presenting cell type are dendritic cells. Despite their important role in EAU, dendritic cells have yet to be fully investigated in human non-infectious uveitis (*Chen et al., 2015b*; *Fu et al., 2019*; *Wang et al., 2021*). CD1c-positive conventional dendritic cells (CD1c+ DCs) have been found to be associated with disease activity (*Chen et al., 2016*; *Chen et al., 2015a*; *Chen et al., 2014*), and are abundant in eye fluid of patients (*O'Rourke et al., 2018*). In order to understand the role of CD1c+ DCs in non-infectious uveitis, it is necessary to understand their functions.

Several single-cell studies have revealed that the CD1c+ DCs (and its murine equivalent, termed 'cDC2s') consists of multiple subsets derived from distinct progenitors (*Villani et al., 2017*; *Dutertre et al., 2019*; *Cytlak et al., 2020*). The type I IFN family of cytokines promotes the expansion of a subset of CD1c+ DCs called 'DC3' (*Dutertre et al., 2019*; *Girard et al., 2020*; *Bourdely et al., 2020*). DC3s are increased in blood of type I IFN-driven *systemic lupus erythematosus* (SLE) patients (*Dutertre et al., 2019*). A significant difference between non-infectious uveitis and SLE is that active uveitis is accompanied by lower levels of type I IFN (*Wang et al., 2019*; *Kuiper et al., 2022*). It should be noted that although type I IFNs can induce lupus-like disease, they can also suppress non-infectious uveitis (*Wang et al., 2019*; *Rönnblom et al., 1991*; *Rönnblom et al., 1990*), pointing to an alternative disease mechanism implicating CD1c+ DCs in non-infectious uveitis. Hence, we do not fully understand the characteristics of CD1c+ DC during autoimmunity, especially in conditions not driven by type I IFNs.

For the purpose of characterizing the core transcriptional features and subset composition of CD1c+ DCs in autoimmunity of the eye, we used whole transcriptome profiling by bulk RNA-sequencing of peripheral blood CD1c+ DCs and multiparameter flow cytometry of two cohorts of non-infectious uveitis patients and healthy donors. We constructed co-expression networks that identified a robust gene module associated with non-infectious uveitis in patients that helped identify a CX3CR1-positive CD1c+ DC subset.

**Table 1.** Characteristics of the patients and controls from cohorts I and II.
Abbreviations: BU: birdshot uveitis, AU: HLA-B27-associated anterior uveitis, HC: healthy control, IU: idiopathic intermediate uveitis, n.a.: not applicable, *Fisher's exact test, **ANOVA, ***Kruskal–Wallis.

| Cohort I | AU | IU | BU | HC | p value |
|---|---|---|---|---|---|
| *N* | 10 | 5 | 8 | 13 | Total = 36 |
| Male/female | 2/8 | 3/2 | 5/3 | 5/8 | 0.26* |
| Age in years; mean ± SD | 45 ± 16 | 30 ± 9 | 42 ± 10 | 42 ± 13 | 0.24*** |
| Disease duration in years; median (range) | 8.1 (0.2–22.3) | 3.4 (0.4–14.1) | 0.9 (0.2–19.9) | n.a. | 0.36** |
| Cohort II | AU | IU | BU | HC | p value |
| *N* | 9 | 9 | 10 | 14 | Total = 42 |
| Male/female | 3/6 | 2/7 | 4/6 | 6/8 | 0.8790* |
| Age in years; mean ± SD | 47 ± 17 | 39 ± 14 | 52 ± 13 | 39 ± 10 | 0.06*** |
| Disease duration in years; median (range) | 5.8 (0.1–39.3) | 3.7 (0.2–20.0) | 1.3 (0.2–15.1) | n.a. | 0.14** |

## Materials and methods

### Patients and patient material

This study was conducted in compliance with the Helsinki principles. Ethical approval was requested and obtained from the Medical Ethical Research Committee in Utrecht (METC protocol number #14-065/M). All patients signed written informed consent before participation. We collected blood from a discovery cohort of 23 and a replication cohort of 28 adult patients (*Table 1*) with HLA-B27-associated acute anterior uveitis (AU), idiopathic intermediate uveitis (IU), or HLA-A29-associated birdshot uveitis (BU). Patients were recruited at the outbound patient clinic of the Department of Ophthalmology of the University Medical Center Utrecht between July 2014 and January 2017. We recruited 27 age- and sex-matched anonymous blood donors of European Ancestry with no history of ocular inflammatory disease at the same institute to serve as unaffected controls (*Table 1*). Uveitis was classified and graded in accordance with the SUN classification (*Jabs et al., 2005*). Each patient underwent a full ophthalmological examination by an ophthalmologist experienced in uveitis, routine laboratory screening, and an X-ray of the lungs. Laboratory screening included erythrocyte sedimentation rate, renal and liver function tests, angiotensin-converting enzyme, and screening for infectious agents in the serum and an Interferon-Gamma Release Assay (IGRA) was obtained for all patients. All patients with AU and BU were HLA-B27 or HLA-A29-positive, respectively (confirmed by HLA typing). All patients had active uveitis (new onset or relapse) and there was no clinical evidence for uveitis-associated systemic inflammatory disease (e.g., rheumatic condition) till the time of sampling. None of the patients received systemic immunomodulatory treatment in the last 3 months, other than low dose (≤10 mg) oral prednisolone in one BU patient of cohort II and one AU patient of cohort I.

### CD1c+ DC purification

Peripheral blood mononuclear cells (PBMCs) were isolated by standard ficoll density gradient centrifugation from 70 ml heparinized blood immediately after blood withdrawal (GE Healthcare, Uppsala, Sweden). For the first cohort, 10 batches (individual days) of 4–5 randomly selected patient and control samples of nitrogen stored PBMCs (mean storage time of 11 [range 0–31] months) were carefully thawed and subjected to sorting by the BD FACSAria III sorter after incubation with a panel of surface antibodies (*Supplementary file 1A*) and fluorescent-activated cell sorting (FACS) buffer (1% bovine serum albumin and 0.1% sodium azide in phosphate-buffered saline [PBS]). CD3−CD19−CD56−CD14−HLA-DR+CD11c+CD1c cells were sorted. The average number of collected cells by sorting was 56,881 (range 6669–243,385). For the second cohort, fresh PBMCs were immediately subjected to magnetic-activated cell sorting (MACS) for the removal (positive selection) of CD304+ cells (pDC), followed by the removal of CD19+ cells (B cell), and subsequently isolation of CD1c+ cells by using the CD1c+ (BDCA1) isolation kit (Miltenyi Biotec, Germany) according to the manufacturer's instructions. The isolated CD1c+ fraction contained on average 147,114 cells (range 46,000–773,000) and purity was determined by flow cytometry (*Supplementary file 1B*) measured on the BD LSRFortessa Cell analyzer (*Figure 1—figure supplement 1*). Data were analyzed using FlowJo software (Tree-Star Inc). MACS or FACS purified CD1c+ cells were immediately taken up in a lysis buffer (RLT plus, QIAGEN) containing 1% β-mercaptoethanol, snap frozen on dry ice, and stored at −80°C until RNA extraction was performed. Isolation of CD1c+ DC for functional experiments was done by MACS as described above. Purification of CD1c+DC subsets based on CD36 and CX3CR1 or CD14 expression from freshly isolated PBMCs was conducted by flow cytometry using the panel in *Supplementary file 1C* and shown in *Figure 3—figure supplement 2B*.

### CD1c+ DC cultures and secretome analysis

Purified CD1c+ DCs were cultured in RPMI Glutamax (Thermo Fisher Scientific) supplemented with 10% heat-inactivated fetal bovine serum (Biowest Riverside) and 1% penicillin/streptomycin (Thermo Fisher Scientific). CD1c+ DCs were cultured at a concentration of 0.5 × 10^6 cells/ml in a 96-well round-bottom plate (100 µl/well). Cells were stimulated overnight (18 hr) with multiple stimuli listed in *Supplementary file 1D*. After stimulation, cells were lysed in an *RLT plus* lysis buffer (QIAGEN) and stored at −80°C until RNA extraction was performed. Cell lysates were stored at −80°C until RNA extraction was performed for qPCR. In separate cultures, CD1c+ DC subsets (sorted based on CD36 and CX3CR1 expression) were cultured in the presence of 1 µg/ml lipoteichoic acid (LTA). After 18 hr of stimulation, supernatants were harvested and IL-23 cytokine production was analyzed by ELISA

(R&D Systems). The levels of IL-2, IL-5, IL-6, IL-10, IL-12p70, IL-13, IL-17, IL-22, IL-27, TNF-alpha, IFN-alpha, IFN-beta, CCL1, CXCL10, CXCL13, VEGF, CD40L, FAS, TNFR1, TNFR2, Elastase, and Granzyme B were simultaneously measured in supernatant of CD1c+ DC cultures using the in-house multiplex immunoassay based on Luminex technology, as described previously (*Bakker et al., 2022*). Protein concentrations that were out of range were replaced with the LLOQ (lower limit of quantification) and ULOQ (upper limit of quantification) for each analyte and divided by 2 for the proteins detected below the range of detection or multiplied by 2 for values above the detection range (*Supplementary file 1E*).

## Real-time quantitative PCR

First-strand cDNA was synthesized from total RNA using Superscript IV kit (Thermo Fisher Scientific), and quantitative real-time PCR (RT-qPCR) was performed on the QuantStudio 12k flex System (Life-Technologies), following the manufacturer's instructions. Sequences of the primers used are listed in *Supplementary file 1F* and the *Key Resource Table*. RT-qPCR data were normalized to the expression of the selected housekeeping gene *GUSB* (ENSG00000169919). CT values were normalized to GUSB by subtracting the CT mean of GUSB (measured in duplo) from the CT mean of the target mRNA (e.g., *RUNX3*) = ΔCT. The fold change (FC) of each sample was calculated compared to ΔCt of the medium control using the formula FC = $2^{-\Delta\Delta Ct}$, where ΔΔCt = ΔCt sample − ΔCt reference.

## Flow cytometry of CD1c+ DC populations

PBMC samples from the two cohorts (HC = 11 samples; AU = 9 samples; IU = 6 samples; BU = 11 samples) were randomly selected and measured by flow cytometry in batches of 9–10 mixed samples per run, divided over 4 days. Per batch, 10 million PBMCs per sample were quickly thawed, washed with ice cold PBS and stained with the antibody panel depicted in *Supplementary file 1C*. PBMCs were incubated with Fixable Viability Dye eF780 (eBioscience) at room temperature for 10 min. Cells were then plated in V-bottomed plates (Greiner Bioone), washed with PBS and incubated for 30 min at 4°C in the dark with Brilliant Stain Buffer (BD) and the fluorescently conjugated antibodies. Next, the cells were washed and taken up in the FACS buffer. Flow cytometric analyses were performed on the BD FACSAria III sorter. Manual gating of data was done using *FlowJo* software (TreeStar inc San Carlos, CA, USA). FlowSOM v1.18.0 analysis was done as described previously (*Laban et al., 2020*). Lineage- (negative for CD3/CD56/CD19) HLA-DR+ data were transformed using the *logicleTransform* function of the *flowCore* v1.52.1 R package, using default parameters (*Ellis et al., 2020*). The SOM was trained for a 7 × 7 grid (49 clusters) with 2000 iterations. Consensus hierarchical clustering was used to annotate clusters, based on the *ConsensusClusterPlus* v1.50.0 R package (*Wilkerson and Hayes, 2010*). Principal component analysis (PCA) analysis was done on normalized expression data from flowSOM using the *factoextra* v 1.0.7.999 R package.

## RNA isolation and RNA-sequencing

Total RNA from CD1c+ DC cell lysates from patients and controls was isolated using the AllPrep Universal Kit (QIAGEN) on the QIAcube (QIAGEN) according to the manufacturer's instructions. For cohort I, RNA-seq libraries were generated by *GenomeScan* (Leiden, The Netherlands) with the TruSeq RNAseq RNA Library Prep Kit (Illumina Inc, Ipswich, MA, USA), and were sequenced using Illumina HiSeq 4000 generating ~20 million 150 bp paired end reads for each sample. Library preparation and Illumina sequencing was performed on samples of cohort II at BGI (Hong Kong). RNA-seq libraries were generated with the TruSeq RNAseq RNA Library Prep Kit (Illumina Inc, Ipswich, MA, USA) and were sequenced using Illumina NextSeq 500 generating approximately 20 million 100 bp paired end reads for each sample.

## Power analysis

We conducted power analysis using the PROPER R package v 1.22.0 (*Wu et al., 2015*) with 100 simulations of the *build-in* RNA-seq count data from antigen presenting (B) cells from a cohort of 41 individuals (i.e., large biological variation as expected in our study) (*Cheung et al., 2010*). Simulation parameters used the default of 20,000 genes and an estimated 10% of genes being differentially expressed. We detected 0.8 power to detect differentially expressed genes ($p < 0.05$) at a $\log_2$(fold

change) >1 for the smallest patient group (9 cases) and we considered the sample size reasonable for analysis.

## Differential gene expression and statistical analysis

Quality check of the raw sequences was performed using the FastQC tool. Reads were aligned to the human genome (GRCh38 build 79) using STAR aligner (*Dobin et al., 2013*) and the Python package HTSeq v0.6.1 was used to count the number of reads overlapping each annotated gene (*Anders et al., 2015*). We aligned the reads of the RNA-sequencing datasets to 65,217 annotated *Ensemble Gene* IDs. Raw count data were fed into *DESeq2* v1.30.1 (*Love et al., 2014*) to identify differentially expressed genes between the four disease groups (AU, IU, BU, and HC). Using DESeq2, we modeled the biological variability and overdispersion in expression data following a negative binomial distribution. We used Wald's test in each disease group versus control pairwise comparison and p values were corrected by the DESeq2 package using the Benjamini–Hochberg Procedure. We constructed co-expression gene networks with the WGCNA v 1.70-3 R package (*Langfelder and Horvath, 2008*) using the cumulative uveitis-associated genes from all pairwise comparisons and a soft power of 5. Module membership (MM) represents the intramodular connectivity of genes in a gene module. Gene Significance (GS >0.25) indicates a strong correlation between genes and non-infectious uveitis, whereas MM (MM >0.8) indicates a strong correlation with the *EigenGene* value of the modules. We calculated the intersection between the modules constructed from the two cohorts and used Fisher's exact test to identify modules that exhibited significant overlap in genes. Gene expression data from *runx3*-knockout (KO) cDC2s, *notch2*-KO cDC2s, and 'inflammatory' cDC2s were obtained from the NCBI Gene Expression Omnibus (accession numbers GSE48590 [2 wild-type [WT] CD11b+ESAM+ splenic cDC2s vs. 2 CD11b+ESAM+ cDC2s from CD11c-DC-*Runx3*Δ mice], GSE119242 [2 untreated cDC2 vs. untreated cDC2 from CD11c-Cre *notch2*f/f mice], GSE149619 [5 CD172+MAR1− cDC2s in mock condition vs 3 CD172+MAR1+ cDC2 in virus condition]) using GEO2R in the GEO database, which builds on the GEOquery v2.58.0 and limma R v 3.46 packages (*Davis and Meltzer, 2007*; *Ritchie et al., 2015*). RNA-seq data from the mouse bone marrow stromal cell line OP9 expressing NOTCH ligand DLL1 (OP9-DLL1)-driven cDC2 cultures (GSE110577 [2 sorted CD11c+MHCII+B220− CD11b+ cDC2 from bone marrow cultures with FLT3L for 7 days vs. 2 sorted CD11c+MHCII+B220− CD11b +cDC2 from bone marrow cultures with FLT3L+OP9-DLL1 cells for 7 days]) were analyzed using DESeq2 and normalized count data plotted using the *plotCounts* function. RNA-seq count data from CD14+/− DC3 subsets from patients with SLE and systemic sclerosis were obtained via GEO (accession number: GSE136731) (*Dutertre et al., 2019*) and differential expression analysis was conducted using *DESeq2* v1.30.1. Gene set enrichment analysis was done using the *fgsea* R package v1.16.0 and data plotted using the *GSEA.barplot* function from the *PPInfer* v 1.16.0 R package (*Jung and Ge, 2020*). Gene sets for *runx3*-KO, *notch2*-KO, inflammatory cDC2s, and cDC2s from OP9-DLL1 bone marrow cultures were generated by taking the top or bottom percentiles of ranked [$-\log_{10}(p) \times sign(\log_2(fold\ change))$] genes from each dataset as indicated. Genes in the modules of interest that encode cell-surface proteins were identified according to *surfaceome* predictor *SURFY* (*Bausch-Fluck et al., 2018*).

## Single-cell RNA-seq analysis of aqueous humor

Single-cell RNA-seq (scRNA-seq) data from a previous study of as reported by *Kasper et al., 2021* of aqueous humor of four HLA-B27-positive anterior uveitis (identical to the AU group in this study) patients were obtained and downloaded via Gene Expression Omnibus (GEO) repository with the accession code GSE178833. Data were processed using the R package *Seurat* v4.1.0 (*Stuart et al., 2019*) using R v4.0.3. We removed low-quality cells (<200 or >2500 genes and mitochondrial percentages <5%) and normalized the data using the *SCTransform*() function accounting for mitochondrial percentage and cell cycle score (*Hafemeister and Satija, 2019*). Dimensionality reduction for all cells was achieved by adapting the original UMAP coordinates for each barcode as reported by *Kasper et al., 2021* (see GSE178833). Data were subjected to *scGate* v1.0.0 (*Andreatta et al., 2022*) using *CLEC10A+* and *C5AR1* (CD88)− cells in our gating model to purify CD1c+ DCs in the scRNAseq dataset. Dimensionality reduction for CD1c+ DCs was conducted using the R package Seurat, and cells clustered using the *FindNeighbors* and *FindClusters* functions from Seurat. After clustering

and visualization with UMAP, we used the *DotPlot* function from the Seurat package to visualize the average expression of genes in each cluster.

## Results

### A CX3CR1 gene module is associated with non-infectious uveitis

We characterized the transcriptome of primary CD1c+ DCs from patients with non-infectious uveitis (*Figure 1A*). RNA-seq analysis (RNA-seq) was performed on lineage (CD3−CD19−CD56−CD14−)-negative, and HLA-DR-positive, CD11c and CD1c-positive DCs purified from frozen PBMCs by flow cytometry from 36 patients with anterior (AU), intermediate (IU), or posterior non-infectious uveitis (BU) and healthy controls. A co-expression network was constructed using uveitis-associated genes identified by differential expression analyses (*n* = 2,016 genes at *P*<0.05, *Figure 1B*), which identified six modules, of which 3 were associated with non-infectious uveitis (GS >0.25, *Figure 1C*). The blue module was most associated with non-infectious uveitis (*Figure 1C*). Based on *Module Membership*, *CX3CR1* was the top hub gene of the blue module (*Figure 1D*, *Supplementary file 1G*). Since CX3CR1 was previously associated with a distinct subset cDC2s that may also express CD14 (*Brown et al., 2019*; *Fujita et al., 2019*), we attempted to validate and expand the gene set associated with non-infectious uveitis by MACS-isolating CD1c+ DC cells from fresh blood of 28 patients and 14 healthy controls, followed by RNA-seq analysis of the highly purified CD1c+ DCs (median [inter-quartile range]% = 96 [3]% pure, *Figure 1—figure supplement 1*). We also constructed a co-expression network for uveitis-associated genes (*n* = 6794, p < 0.05) in the second cohort (*Figure 1E*), which revealed 24 gene modules (*Supplementary file 1H*). Note that patient samples did not cluster according to clinical parameters of disease activity (e.g., cell grade in eye fluid, macular thickness) (*Figure 1—figure supplement 2*). The three uveitis-associated modules in cohort I shared a significant number of co-expressed genes with one module in cohort II, the black module (*Figure 1F*). The black module was associated with non-infectious uveitis in cohort II (GS for uveitis >0.25) and *CX3CR1* was also the hub gene for this module (Black Module Membership, p = 5.9 × 10$^{-22}$; *Supplementary file 1H*, *Figure 1G*). According to these findings, the overlapping disease-associated gene modules appear to represent a single gene module. In cohort I, the separation of genes into three modules was possibly due to low sensitivity to detect disease-associated genes with low expression, as replicated genes of the black module were typically higher expressed (*Figure 1—figure supplement 3*). In total, we replicated 147 co-expressed genes between the two cohorts (which we will refer to as the 'black module'), of which 94% also showed consistent direction of effect (e.g., upregulated in both cohorts) (*Figure 1H*, *Supplementary file 1I*). The black module was enriched for the GO term '*positive regulation of cytokine production*' (GO:0001819, p$_{adj}$ = 6.9 × 10$^{-5}$). In addition to *CX3CR1*, the black module comprised *CD36*, *CCR2*, *TLR-6,-7,-8*, *CD180*, and transcription factors *RUNX3*, *IRF8*, and *NFKB1* (*Figure 1I*), but not *CD14*. In summary, these results show that a gene module characterized by *CX3CR1* in blood CD1c+ DCs is associated with non-infectious uveitis.

### CX3CR1+ DC3 are diminished in peripheral blood of non-infectious uveitis patients

Type I IFN cytokines promote differentiation of CD1c+ DCs (*Dutertre et al., 2019*; *Girard et al., 2020*), but patients with active non-infectious uveitis have reduced blood levels of type I IFN cytokines (*Wang et al., 2019*; *Kuiper et al., 2022*). Assessment of the transcriptome of CD1c+ DCs from patients, found no enrichment for genes associated with murine type I IFN-dependent cDC2s (*Bosteels et al., 2020*; *Figure 2A,B*). Furthermore, while *RUNX3* was downregulated in RNA-seq data from CD1c+ DCs from non-infectious uveitis patients (*Figure 1I*), stimulation of CD1c+ DCs with from healthy human donors with IFN-alpha resulted in upregulation of *RUNX3* (*Figure 2—figure supplement 1A*). In contrast, the transcriptome of CD1c+ DCs from patients overlapped significantly with murine cDC2s knocked out for *Runx3*, or its upstream regulator *Notch2* (*Figure 2*, *Figure 2—figure supplement 1B ,C*; *Lewis et al., 2011*; *Fasnacht et al., 2014*; *Briseño et al., 2018*). Given that cDC2 subsets differ by their dependence on NOTCH signaling (*Lewis et al., 2011*; *Fasnacht et al., 2014*; *Briseño et al., 2018*; *Kirkling et al., 2018*), we hypothesized that the transcriptomic signatures of the CD1c+ DC pool in patients might reflect changes in their proportions.

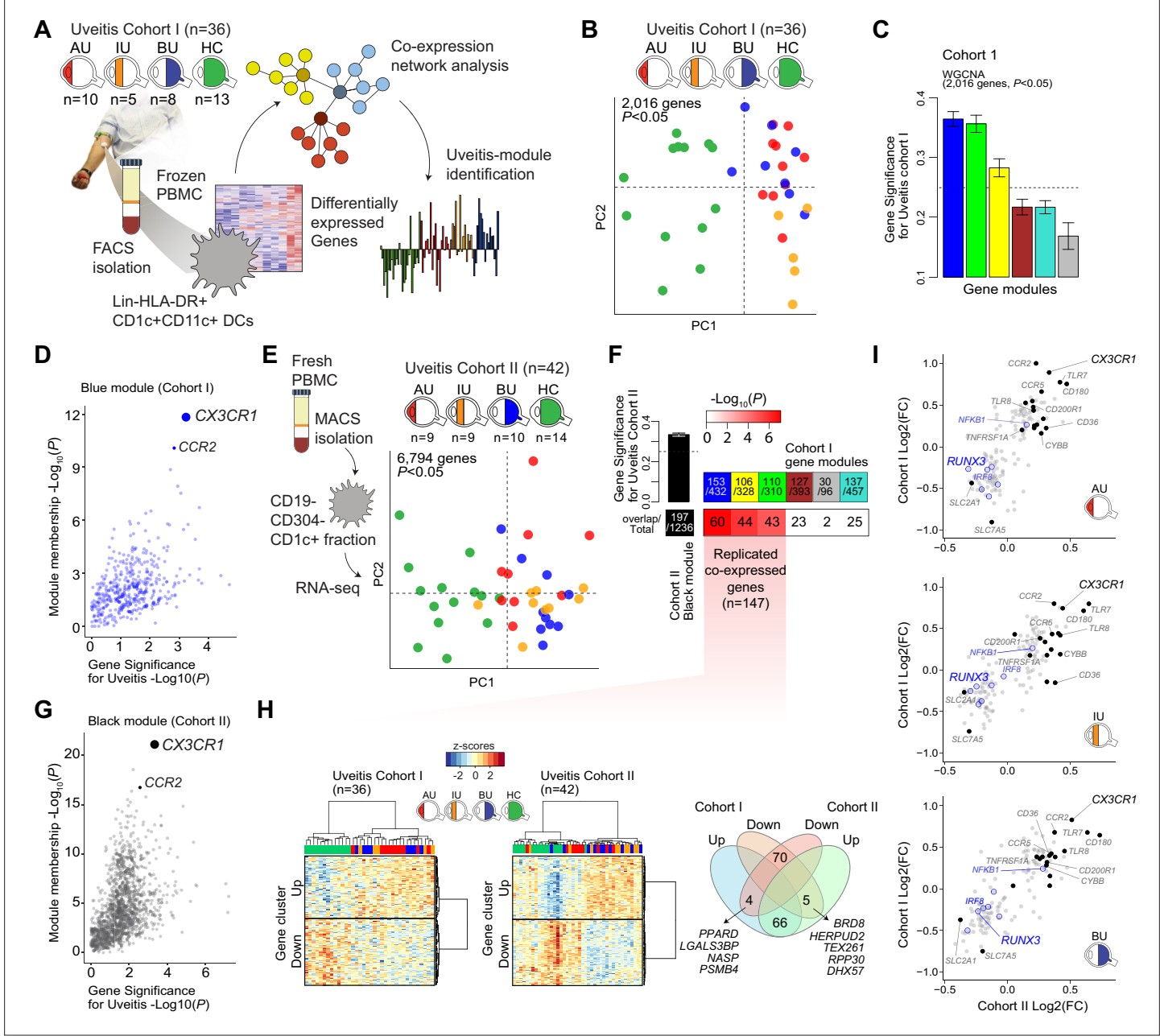

**Figure 1.** A *CX3CR1* gene module in CD1c+ dendritic cells (CD1c+ DC) is associated with non-infectious uveitis. (**A**) Study design. CD1c+ DCs were purified from blood and subjected to RNA-sequencing. Co-expression network analysis was used to identify gene modules associated with uveitis. (**B**) Principal component analysis (PCA) of the 2016 uveitis-associated genes (p < 0.05) in 36 patients and control samples of cohort I. (**C**) Gene significance for uveitis for the gene modules identified by WGCNA. (**D**) Module membership and Gene Significance for uveitis for the blue module of cohort I. (**E**) PCA of the 6794 uveitis-associated genes (p < 0.05) in 42 samples of cohort II. (**F**) Cross-tabulation of the preservation of co-expressed genes between gene modules from cohort I and the black module from cohort II. p value is from Fisher's exact test. (**G**) Same as in *D*, but for the black module of cohort II. (**H**) Heatmaps of the 147 replicated co-expressed genes (rows) for samples (columns) from cohorts I and II. The venn diagram shows the up- and downregulated genes (clusters shown in *H*). (**I**) The (log₂) fold change in gene expression compared to healthy controls (*x*-axis) for all 147 replicated genes in patients with AU, IU, and BU. Genes encoding surface proteins are indicated in black/gray. Key transcription factors are indicated in blue. AU: anterior uveitis, IU: intermediate uveitis, BU: birdshot uveitis.

The online version of this article includes the following figure supplement(s) for figure 1:

**Figure supplement 1.** Purity check of cell fractions for RNA-sequencing in cohort II.

**Figure supplement 2.** Clinical parameters of disease activity in non-infectious uveitis in cohort II.

**Figure supplement 3.** Correlation plot of the mean normalized count (*baseMean* from *DESeq2*) of the black module genes from cohort II and the 147 overlapping genes in the blue, yellow, and green module in cohort I.

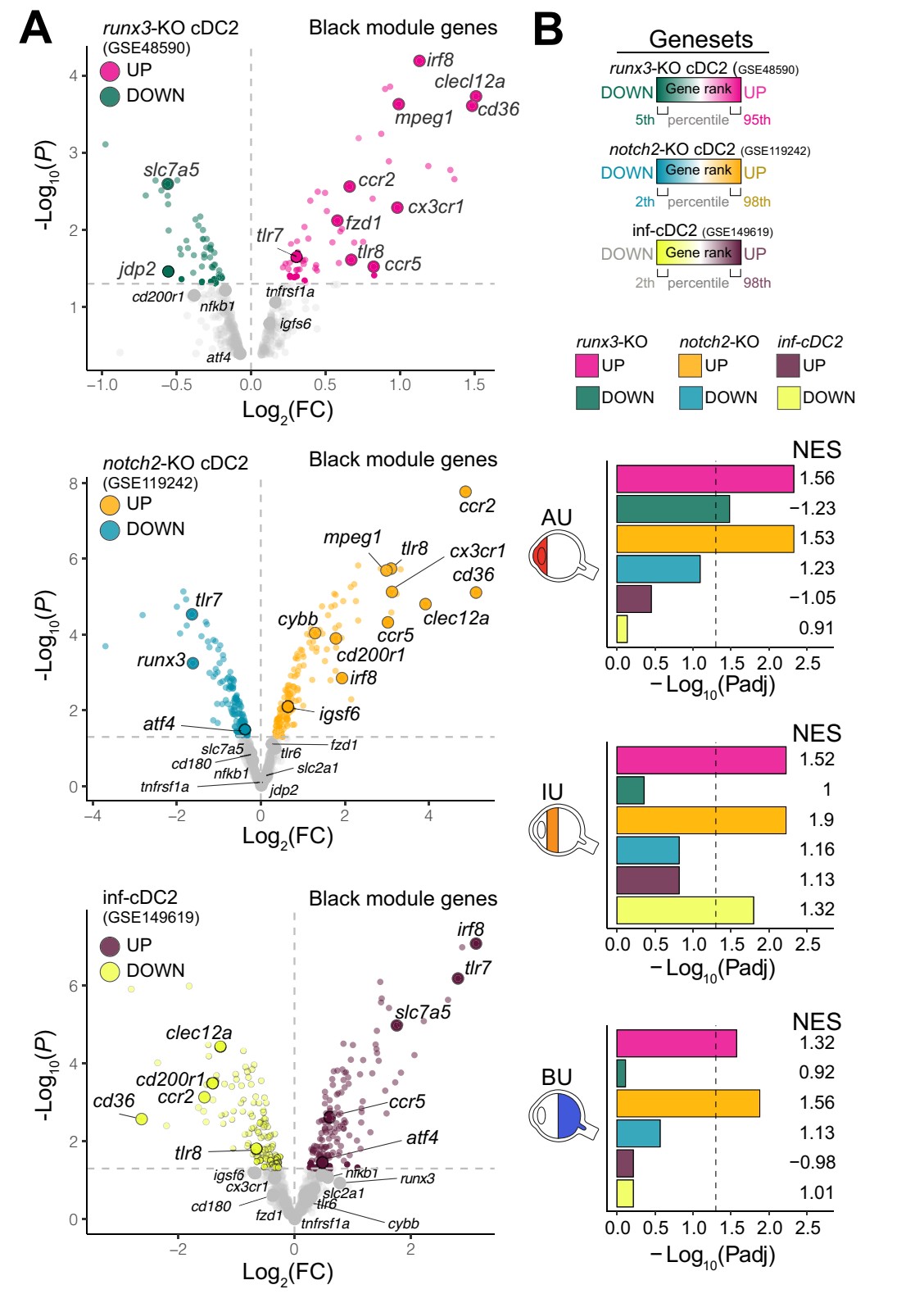

**Figure 2.** The *CX3CR1* gene module of CD1c+ DCs is enriched for NOTCH2-RUNX3 signaling. (**A**) Volcano plot for the expression of genes of the black module in cDC2s of *runx3*-KO mice (GSE48590), *notch2*-KO mice (GSE119242), and type I IFN-dependent inflammatory [inf-]cDC2s (GSE149619). Up- and downregulated genes for each condition are indicated for each condition; gray dots denote the genes with no significant change in expression. (**B**) Results from gene set enrichment analysis for ranked transcriptomes (using 20,668 genes with baseMean >4) for AU, IU, and BU patients. The top or

*Figure 2 continued on next page*

*Figure 2 continued*

bottom percentiles of the ranked [$-\log_{10}(p) \times \text{sign}(\log_2(FC))$] genes from runx3-KO cDC2s, notch2-KO cDC2s, and inf-cDC2s (see a) were used as gene sets. Normalized enrichment scores (NES) and p values for each gene set are indicated. The dotted lines indicate $p_{adj} = 0.05$. AU: anterior uveitis, IU: intermediate uveitis, BU: birdshot uveitis.

The online version of this article includes the following figure supplement(s) for figure 2:

**Figure supplement 1.** In vitro stimulation of CD1c+ DCs and gene enrichment analysis.

Therefore, we used flow-cytometry analysis to identify CD1c+ DC clusters in PBMCs samples from 26 cases and 11 controls. We designed a panel based on the black module (CX3CR1, CD36, CCR2, and CD180), other CD1c+ DC markers that were *not* in the black module (CD1c, CD11c, CD14, CD5, and CD163) (*Dutertre et al., 2019*; *Korenfeld et al., 2017*). FlowSOM (*Van Gassen et al., 2015*) was used on HLA-DR+ and lineage (CD3/CD19/CD56)– PBMCs to cluster cells into a predetermined number of 49 clusters (7 × 7 grid) to facilitate detection of CD1c+ DC phenotypes in blood. The analysis with flowSOM clearly distinguished four CD1c+ DC clusters (cluster number 22, 37, 44, and 45) (*Figure 3A* and *Figure 3—figure supplement 1A and B*). We extracted the data for these four CD1c+ DC clusters and conducted *principal component analysis* (PCA). The PCA biplot identified CD5 and CD163 as top loadings (*Figure 3—figure supplement 1C*), which defines the *DC2s* (cluster 45), CD5− CD163− DC3s (cluster 37), and CD5−CD163+ DC3s (clusters 22 and 44) (*Dutertre et al., 2019*; *Figure 3B, C*). Among the identified clusters, we detected a significant reduction in the frequency of cluster 44 in patients compared to controls (Welch *t*-test, p = 0.03, *Figure 3D* and *Figure 3—figure supplement 1D*). Clusters 44 as well as cluster 22 were CD36 and CD14 positive, which indicates these clusters may represent CD14+ DC3s in human blood (*Dutertre et al., 2019*). However, cluster 44 had relatively higher levels of CX3CR1 than cluster 22 (*Figure 3E, F*). This suggests that DC3s may be phenotypically bifurcated by CX3CR1 independently of CD14. This is supported by weak correlation between *CD14* and *CX3CR1* in our RNA-seq data from bulk CD1c+ DCs (Pearson correlation coefficient = 0.35, *Figure 3—figure supplement 2A*). In addition, we sorted CD14-positive and -negative fractions from CD1c+ DCs of six healthy donors (*Figure 3—figure supplement 2B*) which showed no significant difference in expression levels for CX3CR1 (*Figure 3—figure supplement 2C*). *CX3CR1* levels were also not significantly different in sorted CD5−CD163+CD14-positive and CD14-negative DC3s from patients with autoimmune diseases, further indicating that *CX3CR1* expression in CD1c+ DCs may be independent from *CD14* expression in CD1c+ DCs (*Figure 3—figure supplement 2D*).

We validated by manual gating that cluster 44 represents the CD36+CX3CR1+ fraction of CD1c+ DCs in peripheral blood (~25% of total CD1c+ DCs) (*Figure 3—figure supplement 1E*). Comparison between patients and controls corroborated that the frequency of manual gated CD36+CX3CR1+ DC3s were decreased in the blood of non-infectious uveitis patients (Welch *t*-test, p = 0.029, *Figure 3G*). In detail, we show that CD14+CD1c+ DCs double positive for CD36+ and CX3CR1 were significantly decreased (p = 0.026), while CD14+CD1c+ DCs not positive for CX3CR1 were not (p = 0.43) (*Figure 3G*). This supports that CX3CR1 discerns a phenotypic subpopulation of CD14+ DC3s (*Figure 3—figure supplement 3*), that was diminished in the blood of patients with non-infectious uveitis.

## CX3CR1+ DC3s can secrete pro-inflammatory cytokines upon stimulation

We compared the cytokine-producing abilities of CX3CR1+ DC3s to their negative counterparts since the gene module associated with *CX3CR1* was enriched for genes involved in cytokine regulation. To this end, we freshly sorted primary human CD1c+ DC subsets based on the surface expression of CX3CR1 and CD36, of which double-positive and -negative subsets could be sorted from the selected healthy subjects in sufficient numbers for analysis (*Figure 4—figure supplement 1*). Since CD36 is involved in LTA-induced cytokine production (*Jimenez-Dalmaroni et al., 2009*), we overnight stimulated the CD1c+ subsets with LTA. Both subsets of CD1c+ DCs secreted IL-23 equally strongly (*Figure 4A*). To assess the secretome of the CD1c+ DC subsets in more detail, we profiled the supernatants of LTA-stimulated CD1c+ DC subsets for additional soluble immune mediators (*Supplementary file 1E*): The CD1c+ DC subsets could be distinguished based on the secreted protein profile (*Figure 4B*), of which the levels of TNF-alpha, IL-6, VEGF-A, and TNFR1 showed significant differences

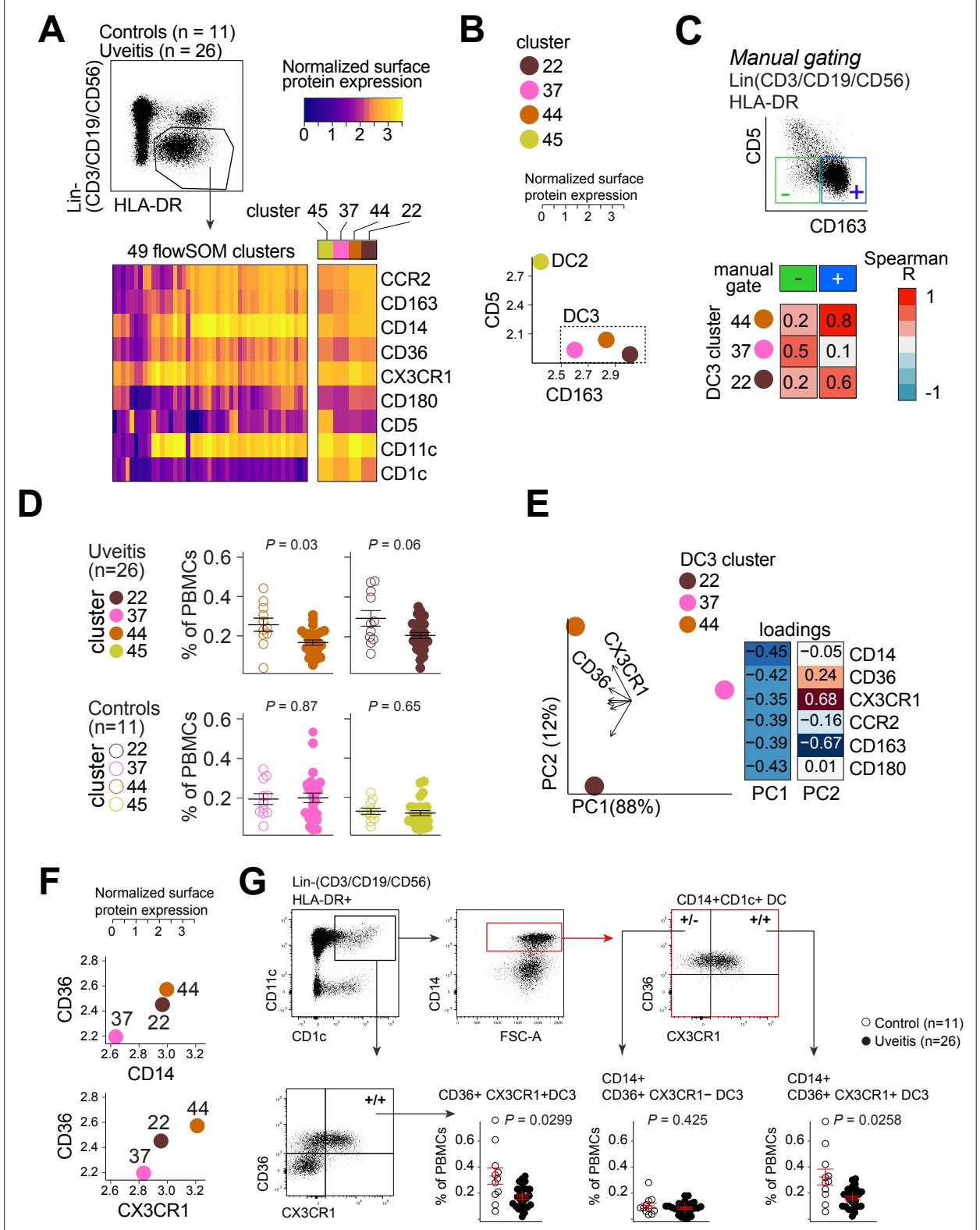

**Figure 3.** CX3CR1+ DC3s are decreased in the blood of patients with non-infectious uveitis. (**A**) Heatmap of the surface protein expression for 49 flowSOM clusters of flow-cytometry analysis of PBMC samples from 26 patients and 11 controls. The four CD1c+ (CD3−CD19−CD56−HLA-DR+CD11c+) DC clusters identified (clusters 22, 37, 44, and 45) are shown (detailed heatmap in *Figure 3—figure supplement 1A*). (**B**) Biplot of the normalized surface expression of CD5 and CD163 for the four CD1c+ DC clusters. (**C**) Correlation plot between manually gated CD5−CD163− DC3s and CD5−

*Figure 3 continued on next page*

*Figure 3 continued*

CD163+ DC3s and DC3 flowSOM clusters 22, 37, and 44. (**D**) The frequency of the 4 CD1c+ DC flowSOM clusters as percentage of PBMCs. p values from Welch's *t*-test. (**E**) Principal component analysis (PCA) biplot of the DC3 clusters 22, 37, and 44. Loadings for PC1 and PC2 are shown on the right. (**F**) Biplots of the normalized surface expression of CD36, CD14, and CX3CR1 in the DC3 clusters 22, 37, and 44. (**G**) Manual gating strategy of CD1c+ DC subsets based on CD36 and CX3CR1 in PBMCs in uveitis cases and controls. p value from Welch's *t*-test. Details on manual gating strategy: see *Figure 3—figure supplement 3*. Manual gating revealed that the CD14+CD1c+ DCs (DC3s) can be further subdivided in a CX3CR1− and a CX3CR1+ population.

The online version of this article includes the following figure supplement(s) for figure 3:

**Figure supplement 1.** Flow cytometry analysis of peripheral blood CD1c+ DC subsets in non-infectious uveitis.

**Figure supplement 2.** Gene expression profiling of CD14+ and CD14- populations of CD1c+ DCs.

**Figure supplement 3.** Representative sample of flow-cytometry gating of CD14+ and CD14− fractions of CD1c+ DCs in peripheral blood for the panel used in *Figure 3*.

between the subsets (*Figure 4C*). These results show that CD1c+ DC subsets defined on the basis of surface co-expression of CD36 and CX3CR1 have the capacity to secrete pro-inflammatory mediators that participate in the pathophysiology of human non-infectious uveitis.

## CX3CR1+ DC3s are detectable in the inflamed eye during non-infectious uveitis

We speculated that CX3CR1+ DC3s are important in the disease mechanisms of uveitis and may be found at increased abundance in the eye during active uveitis. We used single-cell RNA-sequencing data (scRNA-seq) of eye fluid biopsies of four noninfectious patients from *Kasper et al., 2021*. Cells positive for the CD1c+ DC-specific tissue-marker *CLEC10A* and negative for the monocyte marker *C5AR1* (CD88) (*Dutertre et al., 2019*; *Bourdely et al., 2020*; *Heger et al., 2018*) were used to identify CD1c+ DCs among other immune cells in the scRNA-seq data (*Figure 5A*). Unsupervised clustering identified three clusters (1, 2, and 3) of different cells within the CD1c+ DC population (*Figure 5B*). We identified that cluster 1 expressed the gene profile associated with CX3CR1+ DC3s, including relatively higher levels of *CX3CR1*, *CD36*, *CCR2*, and lower levels of *RUNX3* compared to the other two CD1c+ DC clusters (*Figure 5C*), which is in line with the gene profile identified by our bulk RNA-seq analysis. In summary, we conclude that CD1c+ DCs with a gene expression profile similar to CX3CR1+ DC3s can be detected in the eyes of patients during active non-infectious uveitis.

## Discussion

In this study of non-infectious uveitis patients and controls, we identified and replicated a *CX3CR1*-associated gene module in CD1c+ DCs. We were able to track back the gene module to a CX3CR1+ DC3 subset that was diminished in peripheral blood of patients with non-infectious uveitis.

Preceding studies into human CD1c+ DCs revealed functionally distinct subsets termed 'DC2' and 'DC3', with the DC3 showing both transcriptomic features reminiscent of cDC2s and monocytes – such as elevated *CD36* (*Villani et al., 2017*; *Dutertre et al., 2019*). DC3s also have distinct developmental pathways and transcriptional regulators compared to DC2 (*Villani et al., 2017*; *Dutertre et al., 2019*; *Cytlak et al., 2020*; *Bourdely et al., 2020*). Recently, Cytlak et al. revealed that lower expression of *IRF8* is linked to DC3 (*Cytlak et al., 2020*), a transcription factor that was also decreased in non-infectious uveitis. According to *Brown et al., 2019*, CD1c+ DCs exhibit two subsets: cDC2A and cDC2B, whereas cDC2B exhibits higher expression of CX3CR1 and produces more TNF-alpha and IL-6 than cDC2A upon stimulation. Accordingly, uveitis-associated CX3CR1+ DC3s described in this study exhibit similar phenotypical and functional features.

*Dutertre et al., 2019* showed that the phenotype of peripheral blood CD1c+ DCs can be further segregated according to the expression of CD163 and CD5, with 'DC3' cells being characterized as CD5−CD163− or CD5−CD163+ cells and 'DC2' as CD5+CD163 cells. Our flow-cytometry results confirm these findings, but we also show that CD5−CD163+ DC3s that express CD14 are composed of CX3CR1-positive and CX3CR1-negative cells, of which the CX3CR1+ population is implicated in non-infectious uveitis.

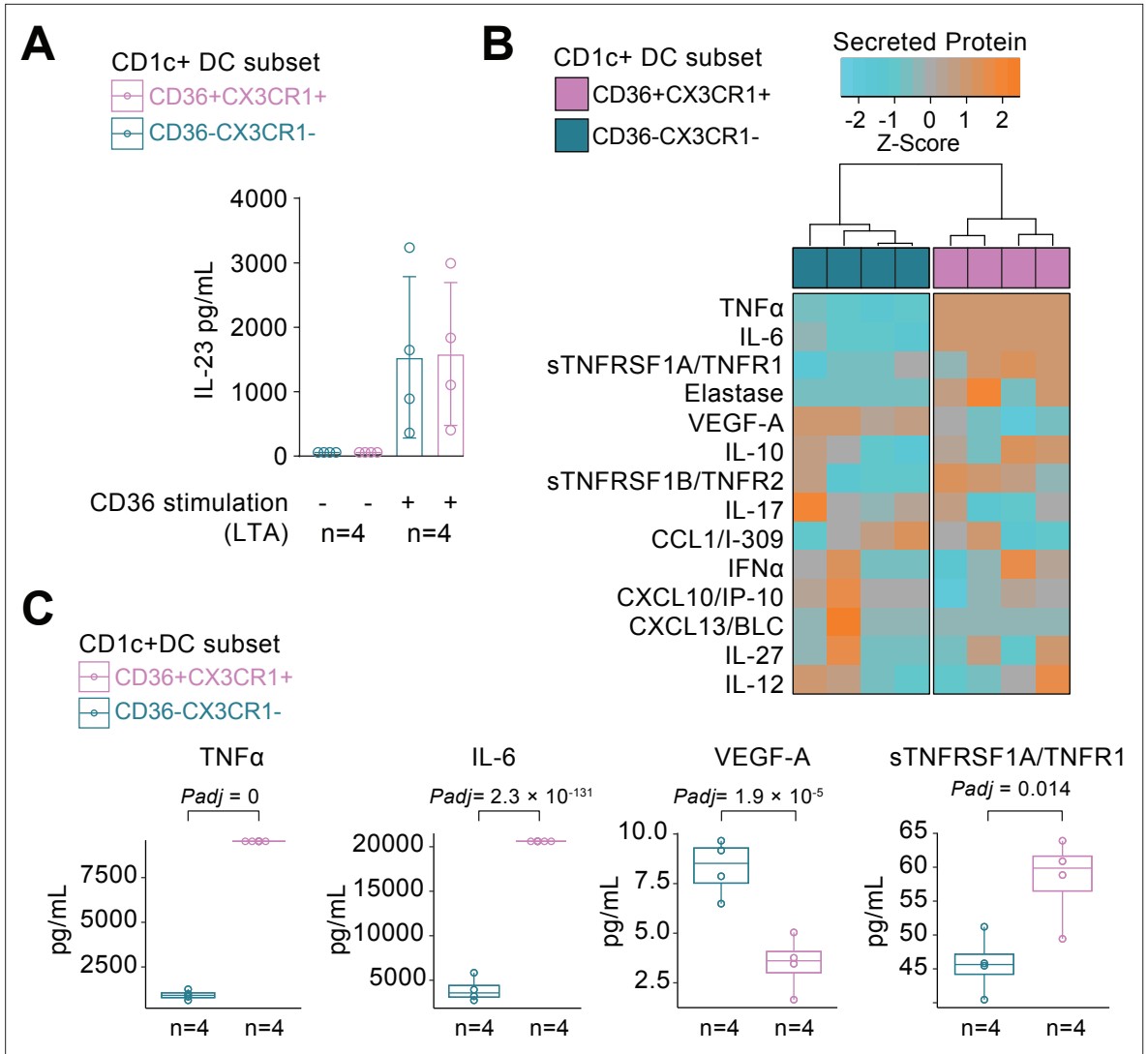

**Figure 4.** CX3CR1+ DC3s secrete high levels of cytokines implicated in non-infectious uveitis. (**A**) The CD1c+ DC cells were fluorescent-activated cell sorting (FACS) sorted into CD36+CX3CR1+ and CD36−CX3CR1−CD1c+ DCs (*Figure 4—figure supplement 1*). The concentration of IL-23 (ELISA) in supernatants of 18 hr cultured primary human CD1c+ DC subsets cells stimulated with lipoteichoic acid (LTA). (**B**) Heatmap of the levels (*Z*-score) of 16 detected proteins in supernatants of 18 hr cultured LTA-stimulated primary human CD1c+ DC subsets cells using an in-house multiplex *Luminex* assay (*Supplementary file 1E*). (**C**) Scatter plots with overlay boxplot with mean and interquartile range of the levels of secreted TNF-alpha, interleukin (IL)-6, VEGF-A, and TNFR1 from the multiplex protein data in *d* ($p_{adj}$ = p values from likelihood ratio test Bonferroni corrected for 16 detected proteins).

The online version of this article includes the following figure supplement(s) for figure 4:

**Figure supplement 1.** Representative examples of fluorescent-activated cell sorting (FACS) of CD36+CX3CR1+ DC3s and CD36−CX3CR1−CD1c+ DCs used for the analysis in *Figure 4*.

Single-cell analysis supported that CD1c+ DCs in eye fluid of patients with non-infectious uveitis contain also a population that has a gene profile reminiscent of CX3CR1+ DC3s, with relatively higher levels of CX3CR1, CD36, CCR2, and lower levels of RUNX3. Patients with SLE display accumulation of CD14+ DC3s in blood (*Dutertre et al., 2019*), while the population of CD14+ DC3 cells was decreased in non-infectious uveitis patients. The differences between non-infectious uveitis and SLE may be related to distinct (i.e., opposite) immunopathological mechanisms; Type I IFNs drive the maturation of cDC2s into 'inflammatory cDC2s' (infcDC2s) (*Bosteels et al., 2020*) and can induce CD1c+ DCs to express a distinct set of surface receptors (*Girard et al., 2020*). The type I IFN-α drives immunopathology of SLE and administration of type I IFN therapy can induce *lupus-like* disease (*Rönnblom et al., 1991*; *Rönnblom et al., 1990*). In favor of attributing the seemingly contrasting

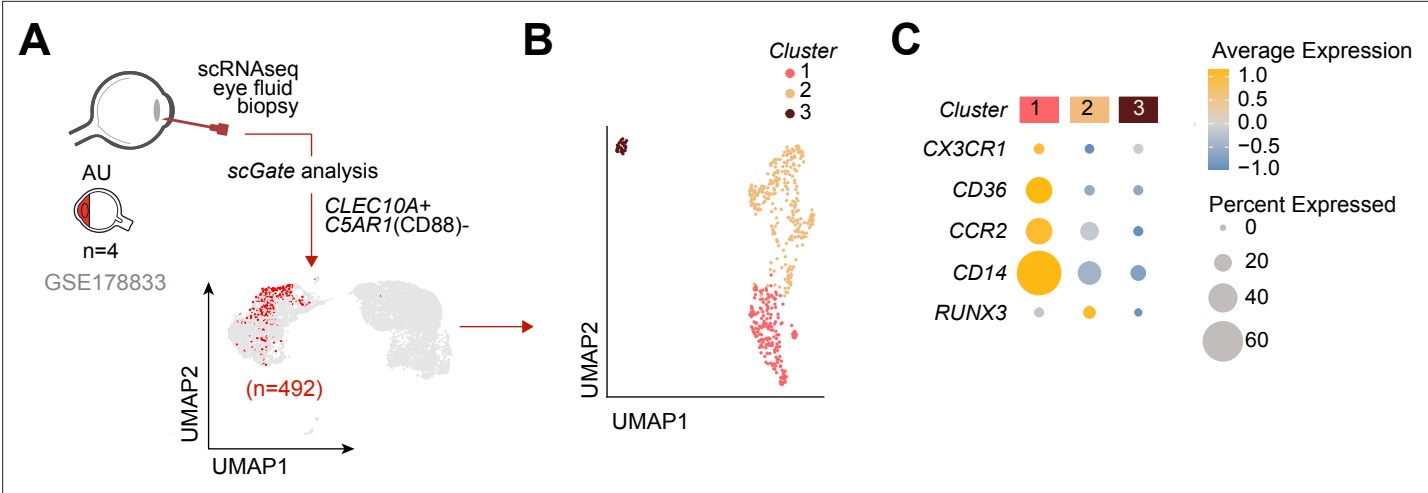

**Figure 5.** Cells with a gene profile similar to CX3CR1+ DC3s can be found in the inflamed eye during non-infectious uveitis. (**A**) Single-cell RNA-sequencing (scRNAseq) analysis of eye fluid biopsies from non-infectious uveitis patients (*GSE178833,* reported by *Kasper et al., 2021*). UMAP projections of transcriptomic data from 492 cells (in red) identified by *scGate* analysis using *CLEC10A+* and *C5AR1–* cells as tissue markers to identify CD1c+ DCs. (**B**) Unsupervised clustering of *CLEC10A+C5AR1–* DCs identified in *a*. (**C**) Dot plot showing average expression (color-scaled) of key marker genes of the black module and *CD14* in each cluster determined in *b*.

observations in blood CD1c+ subsets between SLE and non-infectious uveitis to distinct biology is the fact that, in contrast to elevated IFN-α in patients with SLE, in non-infectious uveitis patient's disease exacerbations correlate with reduced blood type I IFN concentrations (*Wang et al., 2019*; *Kuiper et al., 2022*; *Obermoser and Pascual, 2010*). Despite the importance of type I IFN signaling on DC3s, our results suggest that DC3s are also dysregulated in conditions associated with decreased type I IFNs (*Wang et al., 2019*; *Kuiper et al., 2022*), supporting additional pathways involved in DC3 regulation during chronic inflammation.

We showed that the gene module of CD1c+ DCs showed overlap with the gene signature of disrupted NOTCH2 signaling in cDC2s. *Notch2* signaling is mediated via the NF-κB family member *Relb* in murine cDC2s (*Diener et al., 2021*). NF-κB signaling via RelB suppresses type I IFN signaling in cDC2s (*Saha et al., 2020*) while selective deletion of RelB in dendritic cells protects against autoimmunity (*Diener et al., 2021*; *Andreas et al., 2019*). It is tempting to speculate that the enrichment for NOTCH gene signatures implies altered NF-κB-Relb signaling in CD1c+ DCs, with a mechanism that varies between diseases mediated by type I IFNs (e.g., SLE) and type I IFN-negative diseases (e.g., uveitis). Although this warrants further investigation, some circumstantial evidence for this is the presence of NF-κB family members in the black module, such as *NFKB1* and *NFKBIA*, the former associated previously with a CX3CR1+ cDC2s, while the latter can regulate *Relb* function in dendritic cells (*Shih et al., 2012*; *Sun, 2011*). NF-κB-Relb signaling has been shown to suppress type I IFN via a histone demethylase encoded by *KDM4A* which was also in the black module (*Jin et al., 2014*). Regardless, the NF-κB pathways are regulated by the TNFR1, the main receptor for TNF-alpha (*Hayden and Ghosh, 2014*; *Maney et al., 2014*). TNFR1 is expressed at the cell surface of cDC2s and its ectodomain cleaved by the NOTCH2-pathway regulator ADAM10 (*Maney et al., 2014*; *Yang et al., 2016*; *Iberg et al., 2022*). We showed that both TNF-alpha and sTNFR1 were higher in the secretome of activated CX3CR1+ DC3s. This is in agreement with previous studies on CD36+ DC3s (*Villani et al., 2017*) or CX3CR1+ cDC2B (T-bet–) that also produced higher levels of TNF-alpha (*Brown et al., 2019*). Interestingly, altered NF-κB signaling specifically in cDC2 is associated with clinical response to anti-TNF-alpha therapy (*Andres-Ejarque et al., 2021*). Anti-TNF therapy is effective for treatment of non-infectious uveitis (*Touhami et al., 2019*), while anti-TNF therapy may also result in a dysregulated type I IFN response (*Conrad et al., 2018*) indicating potentially cross regulatory mechanisms via NF-κB signaling and type I IFN signaling affecting cDC2s. More research is needed to resolve the regulatory mechanisms driving CD1+ DC changes in type I IFN-negative inflammation, including non-infectious uveitis. It is also possible that the change in CD1c+ DCs observed in this study results from cytokine-induced precursor emigration or differentiation or that the affected peripheral blood DC3s marked by CX3CR1 are in a precursor or pre-activation state.

Other disease modifying factors possibly affect the CD1c+ DC pool in uveitis patients. In mice, antibiotic treatment to experimentally disturb the microbiota affects a cDC2 subset phenotypically similar to CD1c+ DCs and decreases their frequency in the intestine of mice, which suggests microbiota-dependent signals involved in the maintenance of cDC2 subsets (*Brown et al., 2019*). This is especially interesting in light of growing evidence that microbiota-dependent signals cause autoreactive T cells to trigger uveitis (*Horai et al., 2015*), which makes it tempting to speculate that gut-resident cDC2 subsets contribute to the activation of T cells in uveitis models. Dietary components can influence subsets of intestinal dendritic cells (*Ko et al., 2020*). Regardless, most likely, an ensemble of disease modulating factors is involved. For example, myeloid cytokines, such as GM-CSF, contribute to autoimmunity of the eye (*Croxford et al., 2015*) and GM-CSF has been shown to stimulate the differentiation of human CD1c+ subset from progenitors (*Bourdely et al., 2020*). However, GM-CSF signaling in conventional dendritic cells has a minor role in the inception of EAU (*Bing et al., 2020*). Our data support that stimulation of CD1c+subsets with GM-CSF or TLR ligands does not induce the transcriptional features of CD1c+ DCs during non-infectious uveitis, which is in line with previous observations that support that stimulated cDC2s do not convert from one into another subset (*Bourdely et al., 2020*).

Note that our results of a decreased subset of CD1c+ DCs in non-infectious uveitis are in contrast with previous flow-cytometry reports in non-infectious uveitis (*Chen et al., 2016*; *Chen et al., 2015a*; *Chen et al., 2014*). *Chen et al., 2014* reported an increase of CD1c+ myeloid dendritic cells in non-infectious uveitis. It is important to note, however, that their study did not include the DC marker CD11c, thereby including CD1c+CD11c− populations that do not cluster phenotypically with CD1c+ (CD11c+) DCs (e.g., *cluster 46*, se *Figure 3—figure supplement 1A*), which may explain the differences compared to our study.

Better understanding of the changes in the CD1c+ DC pool during human non-infectious uveitis will help develop strategies to pharmacologically influence putative disease pathways involved at an early disease stage, which may lay the foundation for the design of effective strategies to halt progress toward severe visual complications or blindness. Perhaps targeting CD1c+ DCs may be achieved by dietary (microbiome) strategies and provide relatively safe preventive strategies for non-infectious uveitis.

To conclude, we have found that peripheral blood CD1c+ DCs have a gene module linked to a CX3CR1-positive CD1c+ DC subset implicated in non-infectious uveitis.

## Additional information

### Competing interests

Timothy RDJ Radstake: was a principal investigator in the immune catalyst program of GlaxoSmithKline, which was an independent research program. He did not receive any financial support. Currently, TR is an employee of Abbvie where he holds stock. TR had no part in the design and interpretation of the study results after he started at Abbvie. The other authors declare that no competing interests exist.

### Funding

| Funder | Grant reference number | Author |
|---|---|---|
| UitZicht | #2014-4 | Jonas JW Kuiper |
| UitZicht | #2019-10 | Jonas JW Kuiper |
| UitZicht | #2021-4 | Jonas JW Kuiper |

The funders had no role in study design, data collection, and interpretation, or the decision to submit the work for publication.

### Author contributions

Sanne Hiddingh, Resources, Formal analysis, Validation, Investigation, Methodology, Writing – original draft, Patient inclusions and dendritic cell purifications and dendritic cell cultures; Aridaman Pandit, Formal analysis, Validation, Investigation, Methodology; Fleurieke Verhagen, Conceptualization, Data curation, Formal analysis, Investigation, Methodology, Writing – original draft, Patient inclusions

and dendritic cell purifications; Rianne Rijken, Formal analysis, Methodology; Nila Hendrika Servaas, Resources, Formal analysis, Validation, Writing – review and editing, Dendritic cell cultures; Rina CGK Wichers, Formal analysis, Methodology, Knock-down experiments and dendritic cell cultures; Ninette H ten Dam-van Loon, Data curation, Supervision, Investigation, Methodology, Project administration; Saskia M Imhof, Conceptualization, Funding acquisition, Project administration; Timothy RDJ Radstake, Conceptualization, Resources, Data curation, Funding acquisition, Investigation, Writing – review and editing; Joke H de Boer, Conceptualization, Supervision, Funding acquisition, Writing – original draft, Project administration, Writing – review and editing; Jonas JW Kuiper, Conceptualization, Data curation, Formal analysis, Supervision, Funding acquisition, Validation, Investigation, Visualization, Methodology, Writing – original draft, Writing – review and editing

### Author ORCIDs

Aridaman Pandit ⓘ http://orcid.org/0000-0003-2057-9737
Nila Hendrika Servaas ⓘ http://orcid.org/0000-0002-9825-7554
Jonas JW Kuiper ⓘ http://orcid.org/0000-0002-5370-6395

### Ethics

This study was conducted in compliance with the Helsinki principles. Ethical approval was requested and obtained from the Medical Ethical Research Committee in Utrecht. All patients signed written informed consent before participation (METC protocol number #14-065/M).

### Decision letter and Author response

Decision letter https://doi.org/10.7554/eLife.74913.sa1
Author response https://doi.org/10.7554/eLife.74913.sa2

## Additional files

### Supplementary files

• Transparent reporting form

• Supplementary file 1. Supplementary tables. (A) Antibody panel used for sorting peripheral blood mononuclear cells. (B) Antibody panel used for determination of CD1c+ DC purity after MACS isolation. (C) Antibody panel used for phenotyping cDC2 populations in uveitis patients. (D) Overview of stimuli used for CD1c+ DC stimulations in *Figure 3C*. (E) Luminex analysis supernatant of LTA-stimulated CD1c+ DC sorted fraction (protein levels are in pg/mL). (F) Sequences of primers used for RT-qPCR. (G) Results from differential expression analysis and co-expression network analysis in cohort I. (H) Results from differential expression analysis and co-expression network analysis in cohort II. (I) 147 replicated co-expressed genes for cohort 1 and cohort 2

### Data availability

All raw data and data scripts are available via dataverseNL: https://doi.org/10.34894/9Q0FVO and deposited in NCBI's Gene Expression Omnibus accessible through GEO Series accession numbers GSE195501 and GSE194060.

The following datasets were generated:

| Author(s) | Year | Dataset title | Dataset URL | Database and Identifier |
|---|---|---|---|---|
| Kuiper JJ | 2022 | Whole transcriptome-sequencing of CD1c+ conventional type 2 dendritic cells of human non-infectious uveitis patients [Replication cohort] | http://www.ncbi.nlm.nih.gov/geo/query/acc.cgi?acc=GSE195501 | NCBI Gene Expression Omnibus, GSE195501 |

*Continued on next page*

*Continued*

| Author(s) | Year | Dataset title | Dataset URL | Database and Identifier |
|---|---|---|---|---|
| Kuiper JJ | 2022 | Whole transcriptome-sequencing of CD1c+ conventional type 2 dendritic cells of human non-infectious uveitis patients | http://www.ncbi.nlm.nih.gov/geo/query/acc.cgi?acc=GSE194060 | NCBI Gene Expression Omnibus, GSE194060 |
| Kuiper JW | 2022 | Data and R Scripts for: "Transcriptome network analysis implicates CX3CR1-positive type 3 dendritic cells in non-infectious uveitis" | https://doi.org/10.34894/9Q0FVO | DataverseNL, 10.34894/9Q0FVO |

The following previously published datasets were used:

| Author(s) | Year | Dataset title | Dataset URL | Database and Identifier |
|---|---|---|---|---|
| Dicken J, Mildner A, Leshkowitz D, Touw IP | 2014 | The affect of specific ablation of Runx3 from Esam splenic dendritic cells | https://www.ncbi.nlm.nih.gov/geo/query/acc.cgi?acc=GSE48590 | NCBI Gene Expression Omnibus, GSE48590 |
| Briseño CG, Satpathy AT | 2018 | Trancriptional profile of WT and Notch2 cDC2s after immunization with SRBC | https://www.ncbi.nlm.nih.gov/geo/query/acc.cgi?acc=GSE119242 | NCBI Gene Expression Omnibus, GSE119242 |
| Bosteels C, Neyt K, Vanheerswynghels M, van Helden MJ | 2020 | Inflammatory Type 2 cDCs Acquire Features of cDC1s and Macrophages to Orchestrate Immunity to Respiratory Virus Infection | https://www.ncbi.nlm.nih.gov/geo/query/acc.cgi?acc=GSE149619 | NCBI Gene Expression Omnibus, GSE149619 |
| Kirkling ME, Cytlak U, Lau CM, Lewis KL | 2018 | Notch signaling facilitates in vitro generation of cross-presenting classical dendritic cells | https://www.ncbi.nlm.nih.gov/geo/query/acc.cgi?acc=GSE110577 | NCBI Gene Expression Omnibus, GSE110577 |
| Kasper M, Heming M, Heiligenhaus A, Meyer zu Hörste G | 2021 | Intraocular dendritic cells characterize HLA-B27-associated acute anterior uveitis | https://www.ncbi.nlm.nih.gov/geo/query/acc.cgi?acc=GSE178833 | NCBI Gene Expression Omnibus, GSE178833 |

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

# Appendix 1

## Appendix 1—key resources table

| Reagent type (species) or resource | Designation | Source or reference | Identifiers | Additional information |
|---|---|---|---|---|
| Biological sample (*Homo sapiens*) | Aqueous humor (AqH) | Department of Ophthalmology, University Medical Center Utrecht, Utrecht, The Netherlands | | |
| Biological sample (*Homo sapiens*) | Blood Plasma (EDTA tubes) | Department of Ophthalmology, University Medical Center Utrecht, Utrecht, The Netherlands | | |
| Antibody | Anti-human CD14-FITC Clone: TÜK4 (mouse monoclonal) | Miltenyi | CAT# 130-080-701 | (Dilution:) 1:50 |
| Antibody | Anti-human CCR2-BV421 Clone: K036C2 (mouse monoclonal) | BioLegend | CAT# 357210 | (Dilution:) 1:150 |
| Antibody | Anti-human CD11c-PerCP-Cy5.5 Clone: Bu15 (mouse monoclonal) | BioLegend | CAT# 337210 | (Dilution:) 1:200 |
| Antibody | Anti-human CD123-FITC Clone: 6 H6 (mouse monoclonal) | eBioscience | CAT# 11-1239-42 | (Dilution:) 1:20 |
| Antibody | Anti-human CD14-PerCP-Cy5.5 Clone: HCD14 (mouse monoclonal) | BioLegend | CAT# 325622 | (Dilution:) 1:100 |
| Antibody | Anti-human CD14-BV785 Clone: M5E2 (mouse monoclonal) | BioLegend | CAT# 301840 | (Dilution:) 1:100 |
| Antibody | Anti-human CD141 BDCA3-APC Clone: AD5-14H12 (mouse monoclonal) | Miltenyi | CAT# 130-090-907 | (Dilution:) 1:20 |
| Antibody | Anti-human CD163-BV510 Clone: GHI/61 (mouse monoclonal) | BioLegend | CAT# 333628 | (Dilution:) 1:25 |
| Antibody | Anti-human CD180-PE/Cy7 Clone: MHR73-11 (mouse monoclonal) | BioLegend | CAT# 312910 | (Dilution:) 1:100 |
| Antibody | Anti-human CD19-eF450 Clone: HIB19 (mouse monoclonal) | eBioscience | CAT# 48-0199-42 | (Dilution:) 1:25 |
| Antibody | Anti-human CD19-BV605 Clone: SJ25C1 (mouse monoclonal) | BD | CAT# 562653 | (Dilution:) 1:50 |
| Antibody | Anti-human CD19-AF700 Clone: HIB19 (mouse monoclonal) | eBioscience | CAT# 56-0199-42 | (Dilution:) 1:50 |
| Antibody | Anti-human CD1c-APC Clone: L161 (mouse monoclonal) | eBioscience | CAT# 17-0015-42 | (Dilution sorting cohort 2:) 1:50 |
| Antibody | Anti-human CD1c BDCA1-BV421 Clone: L161 (mouse monoclonal) | BioLegend | CAT# 331526 | (Dilution:) 1:25 |
| Antibody | Anti-human CD1c BDCA1-APC Clone: L161 (mouse monoclonal) | eBioscience | CAT# 17-0015-42 | (Dilution purity check cohort 1:) 1:20 |
| Antibody | Anti-human CD20-PE Clone: 2 H7 (mouse monoclonal) | eBioscience | CAT# 12-0209-42 | (Dilution:) 1:50 |
| Antibody | Anti-human CD3-AF700 Clone: UCHT1 (mouse monoclonal) | BioLegend | CAT# 300424 | (Dilution:) 1:50 |
| Antibody | Anti-human CD304 BDCA4-PE Clone: AD5-17F6 (mouse monoclonal) | Miltenyi | CAT# 130-090-533 | (Dilution:) 1:20 |
| Antibody | Anti-human CD36-PE Clone: CB38 (mouse monoclonal) | BD | CAT# 555455 | (Dilution:) 1:200 |
| Antibody | Anti-human CD4-BV711 Clone: OKT4 (mouse monoclonal) | BioLegend | CAT# 317440 | (Dilution:) 1:50 |
| Antibody | Anti-human CD45-PerCP Clone: HI30 (mouse monoclonal) | BioLegend | CAT# 304026 | (Dilution:) 1:100 |
| Antibody | Anti-human CD5-BB515 Clone: UCHT2 (mouse monoclonal) | BD | CAT# 564647 | (Dilution:) 1:100 |
| Antibody | Anti-human CD56-AF700 Clone: B159 (mouse monoclonal) | BD | CAT# 557919 | (Dilution:) 1:50 |
| Antibody | Anti-human CD8-V500 Clone: RPA-T8 (mouse monoclonal) | BD | CAT# 560774 | (Dilution:) 1:50 |

*Appendix 1 Continued on next page*

*Appendix 1 Continued*

| Reagent type (species) or resource | Designation | Source or reference | Identifiers | Additional information |
|---|---|---|---|---|
| Antibody | Anti-human CX3CR1-PE/Dazzle594 Clone: 2 A9-1 (rat monoclonal) | BioLegend | CAT# 341624 | (Dilution:) 1:100 |
| Antibody | Anti-human HLA-DR-BV605 Clone: G46-6 (mouse monoclonal) | BD | CAT# 562845 | (Dilution:) 1:150 |
| Other | Viability dye and labelling reagent. Live/Dead-APC-eF780 | eBioscience | CAT# 65-0865-14 | (Dilution:) 1:1000 |
| Chemical compound, drug | R848 | Invivogen | CAT# tlrl-r848 | (Concentration:) 1 µg/ml |
| Chemical compound, drug | Lipoteichoic acid (LTA) | Sigma-Aldrich | CAT# L2515-5MG | (Concentration:) 1 µg/ml |
| Chemical compound, drug | Lipopolysaccharide (LPS) | Invivogen | CAT# tlrl-3pelps | (Concentration:) 10 ng/ml |
| Chemical compound, drug | Pam3CSK4 | Invivogen | CAT# tlrl-pms | (Concentration:) 5 µg/ml |
| Chemical compound, drug | oxLDL | Cell Biolabs | CAT# STA-214 | (Concentration:) 50 µg/ml |
| Chemical compound, drug | TGFβ-b2 | R&D Systems | CAT# 302-B2-002/CF | (Concentration:) 100 ng/ml |
| Chemical compound, drug | FLT3L | Cellgenix | CAT# 1415-05 | (Concentration:) 100 ng/ml |
| Chemical compound, drug | TNFα | R&D Systems | CAT# 210-TA-020 | (Concentration:) 100 ng/ml |
| Chemical compound, drug | S100A12 (EN-RAGE) | R&D Systems | CAT# 1052-ER-050 | (Concentration:) 1 µg/ml |
| Chemical compound, drug | IL-4 | R&D Systems | CAT# 204-IL-50 | (Concentration:) 10 ng/ml |
| Chemical compound, drug | IFNα-2a | Cell Sciences | CAT# CRI003B | (Concentration:) 1000 U/ml |
| Chemical compound, drug | GM-CSF | R&D Systems | CAT# 215 GM-500 | (Concentration:) 800 U/ml |
| Commercial assay or kit | Olink Target 96 Immuno-Oncology | Olink | CAT# 95311 | Olink Targeted Proteomics analysis |
| Commercial assay or kit | TruSeq RNA Library Prep Kit | Illumina | CAT#RS-122-2001 | RNA-seq |
| Commercial assay or kit | AllPrep DNA/RNA/miRNA Universal Kit | QIAGEN | CAT# 80224 | RNA isolations |
| Commercial assay or kit | CD1c (BDCA-1)+Dendritic Cell Isolation Kit, human | Miltenyi Biotec | CAT#130-119-475 | isolation of CD1c+ DCs from PBMCs. |
| Commercial assay or kit | CD19 MicroBeads, human | Miltenyi Biotec | CAT#130-050-301 | Depletion of CD19+ B cells from PBMCs |
| Commercial assay or kit | CD304 (BDCA-4/Neuropilin-1) MicroBead Kit, human | Miltenyi Biotec | CAT#130-090-532 | Depletion of plasmacytoid dendritic cells from PBMCs |
| Commercial assay or kit | Human IL-23 Quantikine ELISA Kit | R&D Systems | CAT# D2300B | Quantification of IL-23 in supernatant of CD1c+ DC cultures |
| Recombinant DNA reagent | CD36 FW (Sequence 5′–3′) AAAGAGGTCCTTATACGTACAGAGTTCGT | Integrated DNA Technologies | | qPCR primer |
| Recombinant DNA reagent | CD36 RV (Sequence 5′–3′) AGCCTTCTGTTCCAACTGATAGTGA | Integrated DNA Technologies | | qPCR primer |
| Recombinant DNA reagent | RUNX3 FW (Sequence 5′–3′) CAATGACGAGAACTACTCCGC | Integrated DNA Technologies | | qPCR primer |
| Recombinant DNA reagent | RUNX3 RV (Sequence 5′–3′) GAAGCGAAGGTCGTTGAACC | Integrated DNA Technologies | | qPCR primer |

*Appendix 1 Continued on next page*

*Appendix 1 Continued*

| Reagent type (species) or resource | Designation | Source or reference | Identifiers | Additional information |
|---|---|---|---|---|
| Recombinant DNA reagent | GUSB FW (Sequence 5'–3') CACCAGGGACCATCCAATACC | Integrated DNA Technologies | | qPCR primer |
| Recombinant DNA reagent | GUSB RV (Sequence 5'–3') GCAGTCCAGCGTAGTTGAAAAA | Integrated DNA Technologies | | qPCR primer |
| Recombinant DNA reagent | CCR2 FW (Sequence 5'–3') CCACATCTCGTTCTCGGTTTATC | Integrated DNA Technologies | | qPCR primer |
| Recombinant DNA reagent | CCR2 RV (Sequence 5'–3') CAGGGAGCACCGTAATCATAATC | Integrated DNA Technologies | | qPCR primer |
| Recombinant DNA reagent | CX3CR1 FW (Sequence 5'–3') AGTGTCACCGACATTTACCTCC | Integrated DNA Technologies | | qPCR primer |
| Recombinant DNA reagent | CX3CR1 RV (Sequence 5'–3') AAGGCGGTAGTGAATTTGCAC | Integrated DNA Technologies | | qPCR primer |
| Recombinant DNA reagent | IRF8 FW (Sequence 5'–3') CGACGCGCACCATTCA | Integrated DNA Technologies | | qPCR primer |
| Recombinant DNA reagent | IRF8 RV (Sequence 5'–3') GCTTGCCCCCATAGTAGAAGCT | Integrated DNA Technologies | | qPCR primer |
| Recombinant DNA reagent | TLR7 FW (Sequence 5'–3') CAAGAAAGTTGATGCTATTGGGC | Integrated DNA Technologies | | qPCR primer |
| Recombinant DNA reagent | TLR7 RV (Sequence 5'–3') TGGTTGAAGAGAGCAGAGCA | Integrated DNA Technologies | | qPCR primer |
| Recombinant DNA reagent | CCR5 FW (Sequence 5'–3') TGCTACTCGGGAATCCTAAAAACT | Integrated DNA Technologies | | qPCR primer |
| Recombinant DNA reagent | CCR5 RV (Sequence 5'–3') TTCTGAACTTCTCCCCGACAAA | Integrated DNA Technologies | | qPCR primer |
| Software, algorithm | R Project for Statistical Computing; R version 4.0.3 (2020-10-10) | https://www.r-project.org/ | RRID:SCR_001905 | RNA-seq,flowSOM, statistical analysis |
| Software, algorithm | FlowJo v10.6.1 | BD Biosciences; https://www.flowjo.com/solutions/flowjo | RRID:SCR_008520 | Flow cytometry |
| Software, algorithm | Seurat v3.1.5 | *Stuart et al., 2019*; http://seurat.r-forge.r-project.org/ | RRID:SCR_007322 | scRNA-seq analysis |
| Software, algorithm | DESeq2 v1.30.1 | https://bioconductor.org/packages/release/bioc/html/DESeq2.html | RRID:SCR_015687 | RNA-seq analysis |
| Software, algorithm | WGCNA v 1.70-3 | http://www.genetics.ucla.edu/labs/horvath/CoexpressionNetwork/ | RRID:SCR_003302 | Co-expression network analysis |
| Software, algorithm | flowSOM | https://github.com/SofieVG/FlowSOM, *Van Gassen et al., 2023* | RRID:SCR_016899 | Flowcytometry analysis using a Self-Organizing Map. |
| Software, algorithm | UCell v1.3.1 | https://github.com/carmonalab/UCell, *Andreatta and Carmona, 2023* | | scRNAseq Module score |
| Software, algorithm | scGate v1.0.0 | https://github.com/carmonalab/scGate, *Andreatta et al., 2023* | | Purification of intraocular CD1c+ DCs in scRNAseq data |

