## [Editor Report]

These findings are valuable to ocular immunologists who the study pathophysiologic mechanisms driving inflammation in human uveitis, and for future identification of novel therapeutic targets. The authors convincingly perform high dimensional multi-omic analysis of testing and replication cohorts, followed by characterization of a disease-specific cell type using comparative analysis with previously validated experimental datasets. The analysis will be of particular interest to basic and translational ocular immunologists, as well as dendritic cell biologists.

---

## [Decision Letter]

**Decision letter after peer review:**

Thank you for submitting your article "Whole transcriptome-sequencing and network analysis of CD1c+ human dendritic cells identifies cytokine-secreting subsets linked to type I IFN-negative autoimmunity to the eye" for consideration by *eLife*. Your article has been reviewed by 3 peer reviewers, and the evaluation has been overseen by a Reviewing Editor and Betty Diamond as the Senior Editor. The following individuals involved in review of your submission have agreed to reveal their identity: James Walsh (Reviewer #1).

Essential revisions:

1) Improved informatics analysis to correct for false discovery and stronger correlation with intra-ocular cDC2s, per reviewer #3

2) Revision of claims to have identified a new type of cDC2, reframed to fit this cell state into the current, more rigorously defined, classification of cDC2 subtypes (authors have already cited key papers). This analysis may also benefit from stronger analysis of the transcriptional profile of intraocular (ie tissue state) CD36/CX3CR1+ cDC2s. Conclusions must be tempered given concerns of reviewers #1 and #2

Also, please correct number of references, it was at times impossible to determine which sources were being cited.

*Reviewer #1 (Recommendations for the authors):*

It is unclear why the CD14 is plotted vs CD3 in Figure 1F.

Line 166: Repeated word "co-expressed expressed".

Line 247: Run on sentence (and used multiple times).

Line 384: "No minimize bias" needs to be corrected.

*Reviewer #2 (Recommendations for the authors):*

Specific comments:

An issue is that the cd1c+ cells used for RNAseq were isolated using microbeads which is extremely impure. What other cells contaminated the prep, can the gene expression be reliably de-convoluted? What is the cell purity prior to RNAseq?

5B and D how many patients are displayed? Patient numbers should be shown for each figure throughout the manuscript.

Phenotype of the DCs should be shown in the presence of *Notch2* and ADAM10 inhibitor? Is the DC3 reduced? CX3CR1, CD36, ccr2 CD163?

What is the output of the CX3CR1, CD36, ccr2 CD163 DC3 in Notch ligand-conditioing human CD34 HPC derived DCS?

Functional analysis for understanding T cell priming by the new DC3 subsets should be shown. what is the implication of the results for autoimmune Uveitis.

Technical details that should be addressed:

– All versions of the software should be noted in the methods section, for example ConsensusClusterPlus, Deseq2, WGCNA, etc

– For logicleTransform – were the default settings used or were the parameters altered, this needs to be indicated for interpretation of the log linear transformation.

– Heatmap color palettes are not ideal for colorblind readers, would recommend using viridis palettes

– For the differential gene expression and generation of weighted gene co-expression networks, why was the raw p-value used as a cut-off and not adjusted p-value? Was there any controls for possible false positives?

– For the FlowSOM clusters – what are the approximate sizes Cluster 81, 41, 61, and 83? What percent of the peripheral blood? What is the relative proportion of these clusters in Uveitis vs control?

Author should cite Korenfeld at al JCI insight 2017 on page 9

*Reviewer #3 (Recommendations for the authors):*

In general, figure panels should be made larger; several -- particularly in Figure 3 -- have panels that are nearly unreadable even when printed at full-page scale.

[Editors' note: further revisions were suggested prior to acceptance, as described below.]

Thank you for resubmitting your work entitled "Transcriptome network analysis of human CD1c+ dendritic cells identifies an inflammatory cytokine-secreting subpopulation within the CD14+ DC3s that accumulates locally in type I IFN-negative autoimmunity to the eye" for further consideration by *eLife*. Your revised article has been evaluated by Betty Diamond (Senior Editor) and a Reviewing Editor.

The manuscript has been improved but there are some remaining issues that need to be addressed, as outlined below:

The authors utilized deep transcriptional profiling of isolated CD1c+ peripheral blood cells to identify a peripheral blood biomarker of uveitis, followed by flow cytometric analysis of protein expression to test the hypothesis that a subset of CD1c+ cells are reduced in uveitis patients. Identification of peripheral blood biomarkers of uveitis is important and their study is based on analysis of a reasonable number of study subjects with active disease, therefore the data supporting the first conclusion that a differential gene expression profile exists in uveitis is well-supported and convincing. The subsequent analysis which attempts to define and new cell type based on comparison with published expression datasets, while hypothesis-generating, is inadequate, as sufficient phenotypic and functional analyses were not performed to reach their conclusions. Similarly, the comparison with genes expressed by ocular CD1c+ cells from a separate dataset is not sufficient to conclude that the peripheral blood CD1c+ cells migrate into the eye.

*Reviewer #1 (Recommendations for the authors):*

The authors have addressed many of the concerns in the original reviews, and while the discovery of a subpopulation of blood-derived DCs that are changed in uveitis would be interesting the revision does not leave me convinced that this is truly a unique population of cells. Specifically the flow cytometry data in Figure 5/supplement suggests that the CD14 population is derived from a population that is bisected by the original gate, rather than from a population of CD1c+ DCs limiting functional analysis, and the scRNA seq does not show that the cells expressing black module genes are distinct from those that don't.

*Reviewer #2 (Recommendations for the authors):*

The authors set out to utilize advance informatics techniques to advance our understanding of a cell type previously shown to play a role in uveitis, type 2 conventional dendritic cells (previously termed mDC1). They accumulated a valuable set of samples from untreated, active disease and employed a testing and validation cohort which reproduced a core set of genes differentially expressed within the CD1c+ dendritic cells in the peripheral blood of patients with uveitis. Importantly these cells appear altered regardless of the subtype of uveitis. They then astutely question whether the transcriptional signature simply represents different proportions of cell subtypes or states and test this hypothesis using flow cytometry.

They also attempt to probe the mechanism driving this particular cell type/state by cleverly drawing on published data sets, however these hypothesis-generating experiments are not validated experimentally in a uveitis system.

They utilize in vitro analysis of similar cells isolated from human patients and show that the cells can be induced to make a specific set of cytokines which is different from a related dendritic cell type, however the lack of concordance with published activated DC2s, or the transcriptional signature in their own ex vivo activated DC2s raises more questions than it answers.

The manuscript was difficult to read, largely because it is trying to accomplish too many goals, but also because the informatics techniques and external data sets were not described sufficiently for an average reader to readily evaluate the method and conclusions, and the conclusions were overstated throughout the paper. Overall, too many hypotheses were tested, alternative hypotheses were not considered/discussed. The data should probably be divided into at least 2 papers, one which explores the peripheral blood subsets more concretely, and one which attempts to elucidate a mechanisms by which low RUNX3 expression is associated with the genes expressed more highly in DC2s in the peripheral blood of uveitis patients.

1. The rationale for depleting CD14 in the validation cohort is not clear and justified. CD14 expression on dendritic cells has been established in the literature (ex Duterte Immunity 2019). This reviewer wonders if a better flow of data would be to start with the current second cohort, CD14 depleted, ID the black module, realize that some genes discovered have been associated with CD14+DC3, then utilize a second cohort that includes CD14+CD1c+ cells to validate and expand the original gene set.

2. Experiments probing the transcriptional regulation of the uveitis-enriched gene set are not definitive, but hypothesis-generating, and left in this story, are distracting from the primary observation of a cell type differentially present in the blood of uveitis patients.

3. The authors claim to have found an inflammatory cell type, but the gene expression profile does not recapitulate inflammation-induced DC2 gene expression profile. Figure 3 attempts to shed mechanistic light but only opens more questions, like if viral infection in mice (figure 3A lower panel) and multiple inflammatory stimuli (figure 3C) induce a transcriptional program opposite what they are finding in these cells in the peripheral blood of uveitis patients, how are the cells identified in this paper likely relevant to eye inflammation?

4. Pro-inflammatory CD14+DC3 increased in blood in SLE patients, but according to authors interpretation, they find a similar cell type is decreased in peripheral blood in uveitis. The authors adhere to a possible explanation that these cells are trafficking to the eye, but by the same logic, SLE patients should have an even more significant reduction, not increase in their peripheral DC3 counts. This is not discussed.

5. Even more importantly, the Chen et al. papers which are cited generally by the authors, showed the opposite- that CD1c+ DCs are increased in the blood of patients with uveitis, and correlate with disease activity. No attempt is made by the authors to compare their data with this data derived from a larger number of patients, or discuss the difference. The authors have essentially ignored this key discrepancy with previously published data.

6. Despite changing some text around the hypothesis that this cell type is reduced in the blood because it migrates to the eye (numerically impossible as pointed out by prior reviewer #1), this notion is reiterated several other places in the paper and should be removed, and frankly, reconsidered. It is also possible that the change in peripheral blood cell type/state frequency results from cytokine-induced precursor emigration/differentiation. Perhaps there is actually a fraction of DC2-type cells that are increased in the blood in uveitis (as Chen found) but express the transcriptional program identified in the paper because they are pre-activated cells?

7. The analysis of ocular DC2s which concludes that the current cell type, CD14+DC3s with black module gene expression, are present in the eye is not convincing. The methods are not clearly explained, however it appears that rather than utilizing unbiased cluster analysis and performing differential gene expression analysis between the patients with non-infectious uveitis, and the control (endophthalmitis), or using another technique like GSEA which they used in previous figures to correlate the black module genes with the genes expressed by the ocular DC2s, the authors appear to have selected individual cells that expressed the black modules genes, then compared the expression of black module genes to the cells that did not express black module genes.

8. The authors state in their response to reviewer #1 comments "we profiled available eye fluid biopsies and paired plasma by Olink proteomics to measure immune mediators from patients and controls from this study (and several additional samples, including aqueous humor from non-inflammatory cataract controls – see revised Figure 5 panel D). This analysis shows that cytokines produced by CD36+CX3CR1+ DCs such as TNF-α and IL-6 are specifically increased in eye tissue of patients, but not in blood."-Neither this data, or discussion of it are included in the revised manuscript.

Finally, a concern is that the title is a gross overstatement of their findings:

1. They have not demonstrated that the cells in this paper induce inflammation, and especially not in the context of uveitis- only that similar cells from healthy patients produce a different set of cytokines when stimulated in vitro compared to another cell type.

2. They have not demonstrated that these very cells migrate to the eye, only that similar genes are present in a possibly similar ocular cell type in another data set.

3. They do not demonstrate type I IFN-negative autoimmunity in the eye. This was a huge stretch, presumably from prior assumptions about the mechanisms driving uveitis along with the finding that their cell type does not share a transcriptional program with murine DC2s activated in a viral infection.

In regard to addressing the Previous Editor Concerns:

1. The informatics analysis for most of the paper is likely sufficient, however the methods are not communicated succinctly and clearly such that non-informatics experts can understand the rationale and method for each analysis. The analysis of the intraocular DCs was not clear and from the details provided, did appear appropriate.

2. Revision of claim to have identified a new type of cDC2- This is still not satisfactory as:

a. authors have not ruled out the possibility that the cells are monocyte-derived by transcriptional analysis, protein expression or functional analysis. The use of CD14-deplete cells to recapitulate the gene expression profile is not sufficient to determine that the cells in this paper are not monocyte-derived, as CD14-expression is demonstrated on cells confirmed by FLT3L response to be dendritic cells in Duterte et al. Immunity 2019.

b. To be defined as a new subset of the previously defined DC3 subset, one would need to exactly replicate the marker expression and then show that the new markers subset that subset further, the current manuscript may simple be focusing on different genes/proteins expressed by one or more previously described subsets.

c. As this paper is useful for describing a cell type or state that differentiates uveitis from healthy patients, these experiments do not need to be done to publish this paper, but the naming of the cell type should be tempered to simply describe the markers that were expressed and suggest how they fit into the Duterte/Villani schema of DC2/DC3 classification. In actuality, the discrimination of cDC2 from monocyte-derived DC2-like cells has proven difficult in many papers, thus the authors are advised to stay out of the mud, so-to-speak.

[Editors' note: further revisions were suggested prior to acceptance, as described below.]

Thank you for resubmitting your work entitled "Transcriptome network analysis implicates CX3CR1-positive type 3 dendritic cells in non-infectious uveitis" for further consideration by *eLife*. Your revised article has been evaluated by Betty Diamond (Senior Editor) and a Reviewing Editor.

Summary:

CD1c+ dendritic cells have been found in the peripheral blood and eyes of patients with active uveitis. The authors set out to characterize CD1c+ dendritic cells in uveitis. They establish that the CD1c+ population varies between uveitis and healthy controls in both gene expression and the frequency of a subset of CD1c+ DCs, recently termed DC3s. Finally, the authors utilize a previously published dataset to show that cells with similar gene expression can be found in the eye during active uveitis.

Review:

The authors compare sorted CD1c+ DCs from patients with non-infectious uveitis and healthy controls and find a gene expression signature associated with uveitis, regardless of anatomic subtype or severity, that includes expression of the chemokine receptor CX3CR1. They corroborate this finding with a second cohort of CD1c+CD14+ cells, which strengthens the uveitis-specific CD1c+ DC signature.

The authors then compare the genes enriched in these uveitis CD1c+ DCs with previously published datasets analyzing murine CD1c+ DCs. They found more overlap between uveitis patient CD1c+ DC genes and murine RUNX3/*NOTCH2* KO CD1c+ DCs than with murine viral-infected cells. While the data suggests that IFN signaling may be less relevant in these cells, the authors' conclusion that the genes differentially expressed by peripheral blood CD1c+ DCs in uveitis are not mediated by type I IFNs is overstated and alternative explanations should also be considered.

Next, the authors used flow cytometry to show that blood CD36+CX3CR1+CD1c+ DCs (thus labeled DC3s) were diminished in uveitis vs healthy controls, suggesting that the difference in CD1c+ gene expression between uveitis and healthy controls may actually be due to differential presence of CD1c+ subsets. The difference is small, but statistically significant, although the observation could have been strengthened by quantifying this cell type longitudinally in the same patients during active and inactive disease.

Next the authors found that LTA-stimulated CX3CR1+ DC3s from healthy controls secrete higher levels of uveitis-relevant inflammatory cytokines, including TNF-α, compared to CX3CR1- DC3s. This experiment was performed on a small number of healthy controls and not compared with cytokine production by DC3s from uveitis patients, which could have further supported the authors conclusion that the differential gene expression identified in Figure 1 was due to reduced proportions of CX3CR1+ DC3 cells in uveitis patients vs healthy controls, rather than qualitative differences between uveitis and healthy DC3s.

Finally, the authors find expression of CD36 and CX3CR1 on CLEC10A+ (which they use as a proxy for CD1C) cells by aqueous dendritic cells from a previously published dataset, suggesting that DC3s similar to those found at reduced frequency in the peripheral blood are present in aqueous inflammation, supporting their relevance in uveitis.

The manuscript has been improved but there are some remaining issues that need to be addressed, as outlined below:

Specifically, there are errors in the methods, results, and legends that must be corrected.

*Reviewer #1 (Recommendations for the authors):*

In the revised paper entitled "Transcriptome network analysis implicates CX3CR1-positive type 3 dendritic cells in non-infectious uveitis," Hiddingh, Vandit, Verhagen et al. explore the gene expression of PBMCs in non-infectious uveitis patients and demonstrate a CX3CR1-positive Cd1c+ gene signature that is altered in non-infectious uveitis. They show that this population is decreased in the peripheral blood, could be regulated by notch signaling, expresses proinflammatory cytokines upon stimulation with LTA, and are present in the eye during uveitis.

The hypothesis that a cD1c+ population of DCs that could be related to uveitis in humans is intriguing and deserves further study. Their use of multiple methods to explore this population and the use of multiple cohorts are a strength of the manuscript, and it raises many intriguing questions that are potentially interesting, such as if this population is expressing inflammatory cytokines upon stimulation, why are they decreased in uveitis?

There are still areas where the manuscript is hard to follow and there are some concerns with the experimental methodology. The difficulty following the author's story is partially due to errors in cross-referencing statements in the text with their figures that support the data and understanding the experiments from the figure legends. For instance, in lines 313-315, the authors state "Furthermore, in CD1c+ DCs from healthy human donors, IFN-α did not induce downregulation of RUNX3 as observed CD1c+ DCs from non-infectious uveitis patients. Figure 2 – Figure 2 supplement 1". Data to support this statement is not found in Figure 2 or Figure 2 supplements (only healthy control data). The Figure 2 Supplement 1 legend references "the notch-negative condition in d" with a d in that figure.

Methodologically, the backgating for the manual gating of CD11c/CD1c suggests that the CD36+CX3CR1+ population is really part of a larger population of CD11c+ cells, raising the question of if this population is too poorly defined in this experimental context. This concern is slightly ameliorated by the appearance of a CD36hiCX3CR1hiCD1c+ population in the unsupervised clustering.

Despite these weaknesses, there is enough strength in using multiple methods and replication with multiple patient cohorts to overcome these concerns and to utilize it as a basis to further explore the functions of this population in uveitis pathogenesis.

*Reviewer #4 (Recommendations for the authors):*

The authors have responded to most of the previous reviews and have generated a more clear and cohesive manuscript.

Additional recommendations:

Figure 1

The text rationale for CD14 separation is confusing, consider omitting it.

A better methodology would have been to repeat analysis with new cohort I followed by validation using new cohort II rather than simply comparing the cohorts, but this reads more clearly and logically than the prior version and the overall conclusions seem valid.

Figure 1 Sup 1 not needed, emphasized the odd methodology sorting "cohort II" for CD14- recommend omitting this from the final version, or using instead Figure 3- Supplement 2 could be moved to the supplement for Figure 1 to explain why black module (from the CD14-sorted cohort) is stronger than then enriched modules from cohort I.

CD14+ CD1c+CD11c+CD36+CXCR3+ DC3s seem to be a subset of CD1c+CD11c+CD36+CXCR3+ DC3s, which may be why there is a stronger gene expression signature black module from cohort II vs the blue and green modules from cohort 1.

The supplemental experimental data shows that sorted DC3s from healthy peripheral blood treated with a variety of inflammatory stimuli upregulate RUNX3. One alternative explanation not discussed by the authors is that peripheral blood DC3s are in a precursor or pre-activation state.

Text: in CD1c+ DCs from healthy human donors, IFN-α did not induce downregulation of RUNX3 as observed in CD1c+ DCs from non-infectious uveitis patients, however supplemental figure 2 only tests CD1c DCs from healthy patients. CD1c+ DCs from uveitis patients were never stimulated with IFN to test whether they downregulate RUNX3 after this stimuli. This textual discussion of the experimental data is misleading.

Sup figure 3 final panel should be G, not H.

Aqueous scRNA samples are listed as obtained from Utrecht in the methods section and should cite the previous dataset.

Data used from prior sources should be more clearly detailed in legends and text. As the paper reads, it appears that the authors did the murine BMDC on the OP9 culture experiment detailed in Sup Figure 2.

Figure 5 image is very misleading – "purify tissue CD1c+ DCs" suggests that cells were purified resulting in the displayed UMAP. CLEC10A and C5AR should both be shown and the label should not state CD1c+ if this expression was not assessed- this is misleading.

---

## [Author Response]

Essential revisions:1) Improved informatics analysis to correct for false discovery and stronger correlation with intra-ocular cDC2s, per reviewer #3

We would like to thank the editor for the time and effort to review our work. As discussed in the response to reviewer #3, we outlined a 3-step strategy to control for false positive findings in our RNA-seq analysis, which includes the use of an independent cohort and replication of gene modules over individual genes (we believe is a more challenging, but also more robust approach rarely conducted in RNA-seq analysis of dendritic cells – especially in the era of single-cell RNA seq). We have provided data analysis including adjusted *P*-values in Figure 1A and Figure 1- Supplement 2F, as well as in the Supplementary File 1H-1J. We also outlined in answer to reviewer #2 that we significantly expanded the analysis of intra-ocular cDC2s in Figure 6. We hope the editor agrees this has now been adequately addressed.

2) Revision of claims to have identified a new type of cDC2, reframed to fit this cell state into the current, more rigorously defined, classification of cDC2 subtypes (authors have already cited key papers). This analysis may also benefit from stronger analysis of the transcriptional profile of intraocular (ie tissue state) CD36/CX3CR1+ cDC2s. Conclusions must be tempered given concerns of reviewers #1 and #2

We have significantly reanalyzed the data presented in Figure 4 to fit the current nomenclature and show that the uveitis-associated cDC2s are a CD14+ DC3 subset most distinguished by CD36 and CX3CR1 (Figure 4, Figure 4 – Supplement 1, Figure 4 – Supplement 2). We have tempered conclusions based on the suggestions of the reviewers and adapted our discussion to include this current nomenclature. We have also substantially extended the transcriptional profile of intraocular CD36+/CX3CR1+ cDC2s by addition of Figure 6. Here, we clustered eye-infiltrating cDC2s according to the expression of the uveitis-associated gene module and show that these cells show relatively higher expression for *CD36*, *CX3CR1*, and lower *RUNX3*, but comparable levels of *CD14* (Figure 6C) – closely corroborating our blood CD1c+ DC analyses. These DC3s were also found at higher frequency in the eye of patients (Figure 6D). We hope the editor agrees we have substantially improved the analysis of intraocular DCs.

Also, please correct number of references, it was at times impossible to determine which sources were being cited.

We have corrected the numbering of references and added references per suggestion of the reviewers.

Reviewer #1 (Recommendations for the authors):It is unclear why the CD14 is plotted vs CD3 in Figure 1F.

We have changed this in revised Figure 1F and plotted CD14 against CD45 and added additional data on gating for all other markers (CD3, CD19, CD1c) used to assess the purity of the cell fractions used for RNA-seq in the revised Figure 1 – Supplement 1. We hope the reviewer considers this to be more appropriate.

Line 166: Repeated word "co-expressed expressed".Line 247: Run on sentence (and used multiple times).Line 384: "No minimize bias" needs to be corrected.

We thank the reviewer for bringing these typos to our attention and have corrected these accordingly.

Reviewer #2 (Recommendations for the authors):Specific comments:An issue is that the cd1c+ cells used for RNAseq were isolated using microbeads which is extremely impure. What other cells contaminated the prep, can the gene expression be reliably de-convoluted? What is the cell purity prior to RNAseq?

We agree that the reviewer that microbeads may provide a challenge in purification of cell subsets. With this in mind, we therefore went to great lengths to attribute the results to CD1c+ DCs. First of all, we used a series of MACS isolations to remove contamination from other cells. For cohort I, fresh PBMCs were immediately subjected to magnetic-activated cell sorting (MACS) for the removal (positive selection) of CD304+ cells (pDC), followed by CD19^+^ cells (B cell), and subsequently isolation of CD1c+ cells by using the CD1c+ (BDCA1) isolation kit (see *Methods*). We now have provided additional experimental data of flow cytometry analysis of the MACS-isolated cell fractions used for RNA-seq of cohort I. As shown in Figure 1—figure supplement 1, we used highly purified CD1c+ cDC2 cell fractions (median [interquartile range] % = 96[3]% pure) with no difference between groups in any of the residual cell populations. We outlined this in the Results section on page 10. As shown in revised Figure 2A-C (see also *reviewer #1* comment 1), we conducted additional experiment to show that the black module genes are not from CD14+ monocytes (not dependent on CD14+ surface expression), but from CD1c+ DCs. Also, in cohort II, we used CD3-CD19-CD56-HLA-DR+CD11c+CD1c+CD14- cells for RNA-seq analysis sorted by flow cytometry. We hope that the reviewer agrees we have provided sufficient additional information to conclude that the results can be attributed to CD1c+ DCs.

5B and D how many patients are displayed? Patient numbers should be shown for each figure throughout the manuscript.

We have added the amount of patients and donors used for each experiment in Figure 1A, B, D, Figure Supplement 1, Figure 2A-D,F, Figure 3C, Figure 4A,D,G,H,I, Figure 4 – Supplement 1, Figure 5A-D, and Figure 6A, D. We hope the reviewer agrees this has now been sufficiently addressed.

Phenotype of the DCs should be shown in the presence of Notch2 and ADAM10 inhibitor? Is the DC3 reduced? CX3CR1, CD36, ccr2 CD163?

As discussed above *Reviewer #1*: question 3. Upon request, we have conducted experiments using the inhibitors and our CD1c+ DC flow cytometry panel, but experienced significant autofluorescence and artifacts (see gating example in Author response image 1) that hampered the correct identification of CD1c+ DCs and their subsets using CD34+ HPC derived DCs. Unfortunately, so far, our attempts have failed to overcome these artifacts. We therefore believe it is appropriate to remove the supplemental figure from the manuscript. Regardless, we show multiple complementary lines of evidence from transcriptomic analysis in transgenic and human cells that strongly link *NOTCH2* signaling to the black module in cDC2s. We address this in more detail in the Discussion section. We hope the reviewer agrees that for the scope of this paper, this has been sufficiently addressed.

**Author response image 1. sa2fig1:** Manual gating example of human CD34-HPC derived DCs shows substantial autofluorescence.

What is the output of the CX3CR1, CD36, ccr2 CD163 DC3 in Notch ligand-conditioing human CD34 HPC derived DCS?Functional analysis for understanding T cell priming by the new DC3 subsets should be shown. what is the implication of the results for autoimmune Uveitis.Technical details that should be addressed:– All versions of the software should be noted in the methods section, for example ConsensusClusterPlus, Deseq2, WGCNA, etc

We have added the versions for software used in the method section and in the Key Resources Table form.

– For logicleTransform – were the default settings used or were the parameters altered, this needs to be indicated for interpretation of the log linear transformation.

Yes, the default settings were used for the *logicleTransform()* function. We have added a detailed *Rmarkdown* to the study *DataverseNL* repository (see Data availability in methods section or https://doi.org/10.34894/9Q0FVO) so readers can detail and reproduce all our analysis.

– Heatmap color palettes are not ideal for colorblind readers, would recommend using viridis palettes

We agree with the reviewer and have changed the heatmap colors to *viridis* palettes in Figure 4 and Figure 4 – Supplement 1.

– For the differential gene expression and generation of weighted gene co-expression networks, why was the raw p-value used as a cut-off and not adjusted p-value? Was there any controls for possible false positives?

Our aim in this study was to use network-based inference of the CD1c+ DC compartment in non-infectious uveitis with the goal to reproduce gene modules (i.e., pathways) over of individuals genes. Although *WGCNA* does not recommends filtering genes by differential expression, we aimed to prioritize genes linked to uveitis while controlling for possible false positives in 3 steps: (1) As a compromise between not filtering genes and using only a strict fraction of differentially expressed genes at *Padj<0.05*, we included genes with raw *P*<0.05 to construct uveitis-relevant gene modules with sufficient resolution. (2) We used an independent cohort in which we constructed modules and tested for overlap in co-expressed genes (i.e., replicate a gene module) (3) that also should show consistent direction of effect (Figure 2G). In our opinion, the strategy to replicate disease-associated gene modules is much more robust as illustrated by corroboration of these data by unbiased flow cytometry data and scRNAseq analysis. Regardless, we have now also added analysis with adjusted *P*-values in the result section (revised Figure 1B, Figure 1 Supplement 2F), and describe that genes with high module membership were also significantly increased in uveitis after correction for multiple testing in the result section on page 11 (*Padj*<0.05, Supplementary File 1H). We hope the reviewer agrees that we have used several complementary strategies to control for false positives in our analysis.

– For the FlowSOM clusters – what are the approximate sizes Cluster 81, 41, 61, and 83? What percent of the peripheral blood? What is the relative proportion of these clusters in Uveitis vs control?

We would like to thank the reviewer for raising these questions. We determined the approximate cluster sizes for cluster 81,41,61, and 83 which revealed that the clusters together represented <1% in the singlet gate (input data for flowSOM). Because identification of rare cell types is challenging for unsupervised gating algorithms, including *flowSOM* (reliance on cellular density which is non optimal for rare population detection), we aimed to improve detection of CD1c+ DC phenotypes by gating out lymphocytes. To this end, we reanalyzed the flowcytometry data of PBMCs of patients and controls with flowSOM using Lin-(CD3-CD19-CD56-) HLA-DR+ gated cells as input. As shown in Figure 4 – Supplement 1 this clearly identified (again) a cluster of 4 CD1c+ DC phenotypes, which we show are highly reminiscent of *DC2* and *DC3* cells (revised Figure 4B). We have updated Figure 4A-F with this significantly improved data and now show that the percentage of one particular cluster (now called cluster 44) is the single cluster that is significantly decreased in blood of patients (revised Figure 4B,C), and importantly, this cluster is distinguished by higher CD36 and CX3CR1 expression but not CD14, in line with our RNA-seq data and scRNAseq analysis (revised Figure 4E, 4F, 4J, Figure 4 – Supplement 2). Finally, we show that cluster 44 can be well identified by manual gating of CD36+CX3CR1+ CD1c+ DCs. We have outlined this extensive reanalysis in the result section on page 15-16.

Author should cite Korenfeld at al JCI insight 2017 on page 9.

We have added the citation at that positioning in the manuscript (which is now page 15).

Reviewer #3 (Recommendations for the authors):In general, figure panels should be made larger; several -- particularly in Figure 3 -- have panels that are nearly unreadable even when printed at full-page scale.

We have increased the panels of the figures and show some heatmap figures in greater detail (Figure 4 – Supplement 1) in the supplemental data.

[Editors' note: further revisions were suggested prior to acceptance, as described below.]

Reviewer #1 (Recommendations for the authors):The authors have addressed many of the concerns in the original reviews, and while the discovery of a subpopulation of blood-derived DCs that are changed in uveitis would be interesting the revision does not leave me convinced that this is truly a unique population of cells. Specifically the flow cytometry data in Figure 5/supplement suggests that the CD14 population is derived from a population that is bisected by the original gate, rather than from a population of CD1c+ DCs limiting functional analysis.

Our sincere thanks go out to the reviewer for providing additional feedback. We think the reviewer may refer to the Figure supplement of Figure 4 (which in this revision is now Figure 3). Flow cytometry analysis in the figure supplement of Figure 3 provide more detail on CD14 expression in the CD11c+CD1c+ DC gate. We have also updated Figure 3G, which shows that CD14+CD1c+ DCs can be subdivided in CX3CR1+ and CX3CR1- cells. The reviewer will hopefully agree that we have adequately addressed this issue.

…and the scRNA seq does not show that the cells expressing black module genes are distinct from those that don't.

We agree that further improvements can be made to our intital approach. Our aim was to demonstrate that inflamed eyes of patients contain cells that express genes associated with CX3CR1+ DC3s. We reanalyzed the single-cell RNA seq data and sorted CD1c+ DCs using *CLEC10A* as a classical cDC2 marker (Heger et al. Front Immunol. 2018), removing monocytes and macrophages using *C5AR1* (CD88- cells, according to Duterte et al. Immunity 2019, Bourdely et al. Immunity 2020). We then used unsupervised clustering to identify gene clusters in the CD1c+ DCs and compared their average gene expression pattern for genes we associated with CX3CR1+ DC3s. Now we describe that cluster 1 exhibits the pattern of expression we found in peripheral blood to be associated with CX3CR1+ DC3. We hope the reviewer agrees that we have now shown that CD1c+ DCs in eye fluid appear to cluster into distinct clusters, one of which is similar to a CX3CR1+ DC3.

Reviewer #2 (Recommendations for the authors):The authors set out to utilize advance informatics techniques to advance our understanding of a cell type previously shown to play a role in uveitis, type 2 conventional dendritic cells (previously termed mDC1). They accumulated a valuable set of samples from untreated, active disease and employed a testing and validation cohort which reproduced a core set of genes differentially expressed within the CD1c+ dendritic cells in the peripheral blood of patients with uveitis. Importantly these cells appear altered regardless of the subtype of uveitis. They then astutely question whether the transcriptional signature simply represents different proportions of cell subtypes or states and test this hypothesis using flow cytometry.They also attempt to probe the mechanism driving this particular cell type/state by cleverly drawing on published data sets, however these hypothesis-generating experiments are not validated experimentally in a uveitis system.They utilize in vitro analysis of similar cells isolated from human patients and show that the cells can be induced to make a specific set of cytokines which is different from a related dendritic cell type, however the lack of concordance with published activated DC2s, or the transcriptional signature in their own ex vivo activated DC2s raises more questions than it answers.

We appreciate the reviewer's constructive comments and additional feedback. The reviewer may have been confused by the description of "in vivo activated" cDC2s, whereas this should have been “inflammatory” cDC2s, which are type I IFN-Dependent murine cDC2 subsets induced by pneumonia viruses in mice. We regret this mistake. Specifically, we wanted to demonstrate that CD1c+ DC transcriptomes of patients with non-infectious uveitis do not match those of these type I IFN dependent cDC2. The signature of patients is also not induced by type I IFN stimulation. We have clarified this in more detail in the result section.

Furthermore, we show that the CX3CR1-positive CD1c+ DCs in this study secreted similar levels of IL-23 and higher levels of TNF α and IL-6 than the CD36-CX3CR1- DC3 subsets after activation. Interestingly, this observation is in agreement with Villani et al. (Science 2017; PMID: 28428369) who observed that stimulated CD1c+CD36+ DC3s (CD1c_B population) produced more TNF α than CD36-DC3s (CD1c_A). Additionally, Brown et al. (Cell 2019) found that CX3CR1+ cDC2s produce more IL-6 and TNF-α than their CX3CR1-negative counterparts. Combined, our data on ex vivo stimulated cDC2s indicate that the secretome of activated DC3s in this study is consistent with previous findings. This has been added to the Discussion section. Hopefully, the reviewer will agree that these points do answer some questions, but further research is needed to fully understand these changes and their relation to the cause of non-infectious uveitis.

The manuscript was difficult to read, largely because it is trying to accomplish too many goals, but also because the informatics techniques and external data sets were not described sufficiently for an average reader to readily evaluate the method and conclusions, and the conclusions were overstated throughout the paper. Overall, too many hypotheses were tested, alternative hypotheses were not considered/discussed. The data should probably be divided into at least 2 papers, one which explores the peripheral blood subsets more concretely, and one which attempts to elucidate a mechanisms by which low RUNX3 expression is associated with the genes expressed more highly in DC2s in the peripheral blood of uveitis patients.

We have rewritten sections to improve readability. We have provided more details on the analysis to accommodate readers without experience in computational analyses. We have moved the bulk of the enrichment analysis to the supplement and merged figures 1 and 2. In our opinion, this should have improved the readability of the manuscript, and we hope the reviewer agrees.

1. The rationale for depleting CD14 in the validation cohort is not clear and justified. CD14 expression on dendritic cells has been established in the literature (ex Duterte Immunity 2019). This reviewer wonders if a better flow of data would be to start with the current second cohort, CD14 depleted, ID the black module, realize that some genes discovered have been associated with CD14+DC3, then utilize a second cohort that includes CD14+CD1c+ cells to validate and expand the original gene set.

Based on the reviewer's recommendation, we started with the second cohort and re-ran all analyses. We have merged figure 1 and 2 and believe this improved the flow of the work. It is our hope that the reviewer will agree that this issue has now been addressed appropriately.

2. Experiments probing the transcriptional regulation of the uveitis-enriched gene set are not definitive, but hypothesis-generating, and left in this story, are distracting from the primary observation of a cell type differentially present in the blood of uveitis patients.

Based on the reviewer's recommendation, we moved non-essential transcriptional regulation parts to the supplement and removed the separate transcriptional regulation paragraph. We believe it is important to show that there is no enrichment for type I IFN signalling, but rather for signaling pathways that distinguish discrete subsets of CD1c+ DCs, which forms the basis for flow cytometry analysis. The Results section now summarizes the potential transcriptional regulation very briefly. Our hope is that the reviewer now considers this issue better taken care of within the Results section.

3. The authors claim to have found an inflammatory cell type, but the gene expression profile does not recapitulate inflammation-induced DC2 gene expression profile. Figure 3 attempts to shed mechanistic light but only opens more questions, like if viral infection in mice (figure 3A lower panel) and multiple inflammatory stimuli (figure 3C) induce a transcriptional program opposite what they are finding in these cells in the peripheral blood of uveitis patients, how are the cells identified in this paper likely relevant to eye inflammation?

CX3CR1+ DC3s produce more TNF α and IL-6, so they were regarded as “inflammatory” by us. However, we now believe that this is somewhat confusing, particularly regarding other subsets previously called "inflammatory". The manuscript has been revised to remove this. Unlike type I IFN, this subset is associated with different transcriptional programming mediated by unknown molecular cues. CX3CR1+ cDC2s display a different gene profile than inflammatory (type I IFN) cDC2s in our study, which is consistent with previous studies in mice (PMID: 34526403). We have outlined in greater detail what we believe drives these cells in non-infectious uveitis in the Discussion section.

4. Pro-inflammatory CD14+DC3 increased in blood in SLE patients, but according to authors interpretation, they find a similar cell type is decreased in peripheral blood in uveitis. The authors adhere to a possible explanation that these cells are trafficking to the eye, but by the same logic, SLE patients should have an even more significant reduction, not increase in their peripheral DC3 counts. This is not discussed.

Our view is similar to the reviewer's that this line of reasoning is paradoxical. In the Discussion section, we have substantially revised the comparison of type I IFNs in SLE and signaling in dendritic cells of patients with non-infectious uveitis. Hopefully, the reviewer finds that the revised discussion adequately addresses this issue.

5. Even more importantly, the Chen et al. papers which are cited generally by the authors, showed the opposite- that CD1c+ DCs are increased in the blood of patients with uveitis, and correlate with disease activity. No attempt is made by the authors to compare their data with this data derived from a larger number of patients, or discuss the difference. The authors have essentially ignored this key discrepancy with previously published data.

We agree with the reviewer that a comparison with Chen et al's previous work is warranted. Chen and colleagues attempted to quantify CD1c+ myeloid dendritic cells in their work, but CD11c wasn't used in their flow cytometry panel. Based on our data, this approach would include quantification of CD1c+ CD11c-negative cells as well (Author response image 2). If we include all CD1c+ populations (regardless of CD11c), we would have an overall increase in "CD1c+ DCs" (Author response image 2) – similar to the conclusion by *Chen et al.* Even when CD11c is not considered in cluster analysis, however, CD1c+ CD11c- cells (cluster 46) do not cluster with our identified CD1c+ DCs, which indicates they may represent a different type of cell. In the Discussion section, we addressed this issue. We hope the reviewer agrees that this discrepancy has now been adequately discussed.

6. Despite changing some text around the hypothesis that this cell type is reduced in the blood because it migrates to the eye (numerically impossible as pointed out by prior reviewer #1), this notion is reiterated several other places in the paper and should be removed, and frankly, reconsidered. It is also possible that the change in peripheral blood cell type/state frequency results from cytokine-induced precursor emigration/differentiation. Perhaps there is actually a fraction of DC2-type cells that are increased in the blood in uveitis (as Chen found) but express the transcriptional program identified in the paper because they are pre-activated cells?

Based on the reviewer's suggestion in the Discussion section, we have adopted a hypothesis that cytokine-induced precursor emigration/differentiation may also contribute to the change in peripheral blood cell type/state frequency. We have removed the hypothesis that these cells migrate to the eye.

7. The analysis of ocular DC2s which concludes that the current cell type, CD14+DC3s with black module gene expression, are present in the eye is not convincing. The methods are not clearly explained, however it appears that rather than utilizing unbiased cluster analysis and performing differential gene expression analysis between the patients with non-infectious uveitis, and the control (endophthalmitis), or using another technique like GSEA which they used in previous figures to correlate the black module genes with the genes expressed by the ocular DC2s, the authors appear to have selected individual cells that expressed the black modules genes, then compared the expression of black module genes to the cells that did not express black module genes.

We agree with the reviewer that a more unbiased cluster analysis of ocular CD1c+ dendritic cells would be more appropriate. Therefore, we reanalyzed the single cell RNA seq data and conducted cluster analysis of the CD1c+ DCs. Because we aimed to demonstrate that inflamed eyes of patients contain cells that express key genes associated with CX3CR1+ DC3s, we sorted single-cell RNA seq data for CD1c+ using CLEC10A as a classical tissue-cDC2 marker (Heger et al. Front Immunol. 2018), removing monocytes using C5AR1 (CD88- cells, according to Duterte et al. Immunity 2019, Bourdely et al. Immunity 2020). We then clustered cells and compared their gene expression for genes associated with CX3CR1+ DC3. Cluster 1 exhibits the pattern of expression we found in peripheral blood to be associated with CX3CR1+ DC3. We hope the reviewer agrees that CD1c+ DCs in eye fluid appear to cluster into distinct clusters, one of which has a gene profile of key genes that is reminiscent of CX3CR1+ DC3s. Hopefully, the reviewer will agree that we have now addressed this section more effectively.

8. The authors state in their response to reviewer #1 comments "we profiled available eye fluid biopsies and paired plasma by Olink proteomics to measure immune mediators from patients and controls from this study (and several additional samples, including aqueous humor from non-inflammatory cataract controls – see revised Figure 5 panel D). This analysis shows that cytokines produced by CD36+CX3CR1+ DCs such as TNF-α and IL-6 are specifically increased in eye tissue of patients, but not in blood."-Neither this data, or discussion of it are included in the revised manuscript.

We regret this mistake in the revision. This description should not have been included in the rebuttal.

Finally, a concern is that the title is a gross overstatement of their findings:1. They have not demonstrated that the cells in this paper induce inflammation, and especially not in the context of uveitis- only that similar cells from healthy patients produce a different set of cytokines when stimulated in vitro compared to another cell type.2. They have not demonstrated that these very cells migrate to the eye, only that similar genes are present in a possibly similar ocular cell type in another data set.3. They do not demonstrate type I IFN-negative autoimmunity in the eye. This was a huge stretch, presumably from prior assumptions about the mechanisms driving uveitis along with the finding that their cell type does not share a transcriptional program with murine DC2s activated in a viral infection.

We agree with the reviewer and have changed the title to: “Transcriptome network analysis implicates CX3CR1-positive type 3 dendritic cells in non-infectious uveitis”. This seems to be more appropriate for the manuscript, and we hope the reviewer agrees.

In regard to addressing the Previous Editor Concerns:1. The informatics analysis for most of the paper is likely sufficient, however the methods are not communicated succinctly and clearly such that non-informatics experts can understand the rationale and method for each analysis. The analysis of the intraocular DCs was not clear and from the details provided, did appear appropriate.

We would like to thank the editor for his/her time. We have amended the single cell RNA sequencing analysis in response to both reviewers to accommodate unbiased clustering and reworked the figures and manuscript as needed.

2. Revision of claim to have identified a new type of cDC2- This is still not satisfactory as:a. authors have not ruled out the possibility that the cells are monocyte-derived by transcriptional analysis, protein expression or functional analysis. The use of CD14-deplete cells to recapitulate the gene expression profile is not sufficient to determine that the cells in this paper are not monocyte-derived, as CD14-expression is demonstrated on cells confirmed by FLT3L response to be dendritic cells in Duterte et al. Immunity 2019.b. To be defined as a new subset of the previously defined DC3 subset, one would need to exactly replicate the marker expression and then show that the new markers subset that subset further, the current manuscript may simple be focusing on different genes/proteins expressed by one or more previously described subsets.c. As this paper is useful for describing a cell type or state that differentiates uveitis from healthy patients, these experiments do not need to be done to publish this paper, but the naming of the cell type should be tempered to simply describe the markers that were expressed and suggest how they fit into the Duterte/Villani schema of DC2/DC3 classification. In actuality, the discrimination of cDC2 from monocyte-derived DC2-like cells has proven difficult in many papers, thus the authors are advised to stay out of the mud, so-to-speak.

Our naming of the population has been changed to CX3CR1+ DC3s, which describes the CD5^-^ CD163+ DC3s cells that are CX3CR1-positive in flow cytometry analysis. As a result, we have removed any description related to discriminating cDC3s from monocyte-derived DC3-like cells. We hope the editors agree that this has now been dealt with more clearly and without making any unsubstantiated claims.

[Editors' note: further revisions were suggested prior to acceptance, as described below.]

Reviewer #1 (Recommendations for the authors):In the revised paper entitled "Transcriptome network analysis implicates CX3CR1-positive type 3 dendritic cells in non-infectious uveitis," Hiddingh, Vandit, Verhagen et al. explore the gene expression of PBMCs in non-infectious uveitis patients and demonstrate a CX3CR1-positive Cd1c+ gene signature that is altered in non-infectious uveitis. They show that this population is decreased in the peripheral blood, could be regulated by notch signaling, expresses proinflammatory cytokines upon stimulation with LTA, and are present in the eye during uveitis.The hypothesis that a cD1c+ population of DCs that could be related to uveitis in humans is intriguing and deserves further study. Their use of multiple methods to explore this population and the use of multiple cohorts are a strength of the manuscript, and it raises many intriguing questions that are potentially interesting, such as if this population is expressing inflammatory cytokines upon stimulation, why are they decreased in uveitis?

We would like to thank the reviewer for his/hers time to review our work and pointing out the strength our our revised work.

There are still areas where the manuscript is hard to follow and there are some concerns with the experimental methodology. The difficulty following the author's story is partially due to errors in cross-referencing statements in the text with their figures that support the data and understanding the experiments from the figure legends. For instance, in lines 313-315, the authors state "Furthermore, in CD1c+ DCs from healthy human donors, IFN-α did not induce downregulation of RUNX3 as observed CD1c+ DCs from non-infectious uveitis patients. Figure 2 – Figure 2 supplement 1". Data to support this statement is not found in Figure 2 or Figure 2 supplements (only healthy control data).

Our apologies for how the sentence in line 313 was formulated. We meant to say that while *RUNX3* was lower in patients with uveitis in RNA-seq data, stimulation of CD1c+ DCs from healthy controls with IFN-α did result in a decrease in *RUNX3* as support for that fact that IFN-α is unlikely causing the lower expression of *RUNX3* as observed by RNA-seq in uveitis patients. It has been changed to “Furthermore, while *RUNX3* was downregulated in RNA-seq data from CD1c+ DCs from non-infectious uveitis patients (Figure 1I), stimulation of CD1c+ DCs with from healthy human donors with IFN-α resulted in upregulation of *RUNX3”.* Our hope is that the reviewer will agree that this is a more appropriate formulation.

The Figure 2 Supplement 1 legend references "the notch-negative condition in d" with a d in that figure.

This was a typo and has been changed to "the notch-negative condition in b".

Methodologically, the backgating for the manual gating of CD11c/CD1c suggests that the CD36+CX3CR1+ population is really part of a larger population of CD11c+ cells, raising the question of if this population is too poorly defined in this experimental context. This concern is slightly ameliorated by the appearance of a CD36hiCX3CR1hiCD1c+ population in the unsupervised clustering.Despite these weaknesses, there is enough strength in using multiple methods and replication with multiple patient cohorts to overcome these concerns and to utilize it as a basis to further explore the functions of this population in uveitis pathogenesis.

We are grateful for the reviewer's time and efforts, and are pleased that he/she believes our work is strong enough to publish.

Reviewer #4 (Recommendations for the authors):The authors have responded to most of the previous reviews and have generated a more clear and cohesive manuscript.

Thanks to the reviewer for reviewing our work and providing recommendations to improve data presentation clarity.

Additional recommendations:Figure 1The text rationale for CD14 separation is confusing, consider omitting it.A better methodology would have been to repeat analysis with new cohort I followed by validation using new cohort II rather than simply comparing the cohorts, but this reads more clearly and logically than the prior version and the overall conclusions seem valid.Figure 1 Sup 1 not needed, emphasized the odd methodology sorting "cohort II" for CD14- recommend omitting this from the final version, or using instead Figure 3- Supplement 2 could be moved to the supplement for Figure 1 to explain why black module (from the CD14-sorted cohort) is stronger than then enriched modules from cohort I.

As recommended by previous reviewers, we have included Figure 1 Supplement 1 and moved the CD14 analysis to Figure 3 – Supplement 2 (instead of near Figure 1). We believe it is important to highlight this section since it emphasises that CD14+ DC3s and CX3CR1+ DC3s are not identical populations. The current structure of the manuscript conforms to previous recommendations from other reviewers. We hope the reviewer agrees.

CD14+ CD1c+CD11c+CD36+CXCR3+ DC3s seem to be a subset of CD1c+CD11c+CD36+CXCR3+ DC3s, which may be why there is a stronger gene expression signature black module from cohort II vs the blue and green modules from cohort 1.

The possibility of such a scenario is not ruled out. However, we show that *CX3CR1* and *CD14* expressions are moderately correlated (Pearson's correlation coefficient = 0.35, Figure 3 Supp 2). Furthermore, Figure 3G shows that only the CD14 DC3 population that expresses CX3CR1 is significantly altered in patients when looking at CD14 DC3s.

The supplemental experimental data shows that sorted DC3s from healthy peripheral blood treated with a variety of inflammatory stimuli upregulate RUNX3. One alternative explanation not discussed by the authors is that peripheral blood DC3s are in a precursor or pre-activation state.

We agree that this may be a possibility. We have added this to the Discussion section as follows “It is also possible that the change in CD1c+ DCs observed in this study results from cytokine-induced precursor emigration or differentiation or that the affected peripheral blood DC3s marked by CX3CR1 are in a precursor or pre-activation state.”

Text: in CD1c+ DCs from healthy human donors, IFN-α did not induce downregulation of RUNX3 as observed in CD1c+ DCs from non-infectious uveitis patients, however supplemental figure 2 only tests CD1c DCs from healthy patients. CD1c+ DCs from uveitis patients were never stimulated with IFN to test whether they downregulate RUNX3 after this stimuli. This textual discussion of the experimental data is misleading.

Our apologies for how the sentence in line 313 was formulated. We meant to say that while *RUNX3* was lower in patients with uveitis in RNA-seq data, stimulation of CD1c+ DCs from healthy controls with IFN-α did result in a decrease in *RUNX3* as support for that fact that IFN-α is unlikely causing the lower expression of *RUNX3* as observed by RNA-seq in uveitis patients. It has been changed to “Furthermore, while *RUNX3* was downregulated in RNA-seq data from CD1c+ DCs from non-infectious uveitis patients (Figure 1I), stimulation of CD1c+ DCs with from healthy human donors with IFN-α resulted in upregulation of *RUNX3”.* Our hope is that the reviewer will agree that this is a more appropriate formulation.

Sup figure 3 final panel should be G, not H.

This was a typo. We have changed the panel figures in the legend of Figure 3 – Supplement 1.

Aqueous scRNA samples are listed as obtained from Utrecht in the methods section and should cite the previous dataset.

We regret that this was not clear for ther reviewer, the method now states that “Single cell RNA-seq (scRNA-seq) data from a previous study of as reported by *Kasper et al. 2021* of aqueous humor of 4 HLA-B27-positive anterior uveitis (identical to the AU group in this study) patients were obtained and downloaded via Gene Expression Omnibus (GEO) repository with the accession code GSE178833.” We hope the reviewer agrees this is now further clarfified in sufficient details.

Data used from prior sources should be more clearly detailed in legends and text. As the paper reads, it appears that the authors did the murine BMDC on the OP9 culture experiment detailed in Sup Figure 2.

Figure legends and text now include more details on the paper references from which we obtained data.

Figure 5 image is very misleading – "purify tissue CD1c+ DCs" suggests that cells were purified resulting in the displayed UMAP. CLEC10A and C5AR should both be shown and the label should not state CD1c+ if this expression was not assessed- this is misleading.

We agree with the reviewer. We now changed this back to the description of using *scGate* analysis to select *CLECL10A+* and *C5AR1*- negative cells in the figure and figure legend. We hope the reviewer agrees this better reflects the analysis conducted.